# Revisiting Multi-Agent World Modeling from a Diffusion-Inspired Perspective

**Yang Zhang[1]\*, Xinran Li[2], Jianing Ye[3], Shuang Qiu[4], Delin Qu[5],**
**Xiu Li[1], Chongjie Zhang[3], Chenjia Bai[6,7]†**

[1]Tsinghua University, [2]The Hong Kong University of Science and Technology,
[3]Washington University in St. Louis, [4]City University of Hong Kong, [5]Fudan University,
[6]Institute of Artificial Intelligence (TeleAI), China Telecom,
[7]Shenzhen Research Institute of Northwestern Polytechnical University
z-yang21@mails.tsinghua.edu.cn, baicj@chinatelecom.cn

## Abstract

World models have recently attracted growing interest in Multi-Agent Reinforcement Learning (MARL) due to their ability to improve sample efficiency for policy learning. However, accurately modeling environments in MARL is challenging due to the exponentially large joint action space and highly uncertain dynamics inherent in multi-agent systems. To address this, we reduce modeling complexity by shifting from jointly modeling the entire state-action transition dynamics to focusing on the state space alone at each timestep through sequential agent modeling. Specifically, our approach enables the model to progressively resolve uncertainty while capturing the structured dependencies among agents, providing a more accurate representation of how agents influence the state. Interestingly, this sequential revelation of agents' actions in a multi-agent system aligns with the reverse process in diffusion models—a class of powerful generative models known for their expressiveness and training stability compared to autoregressive or latent variable models. Leveraging this insight, we develop a flexible and robust world model for MARL using diffusion models. Our method, **D**iffusion-**I**nspired **M**ulti-**A**gent world model (DIMA), achieves state-of-the-art performance across multiple multi-agent control benchmarks, significantly outperforming prior world models in terms of final return and sample efficiency, including MAMuJoCo and Bi-DexHands. DIMA establishes a new paradigm for constructing multi-agent world models, advancing the frontier of MARL research. Codes are open-sourced at https://github.com/breez3young/DIMA.

## 1 Introduction

Learning accurate world models to capture environmental dynamics is crucial for effective decision-making. In the realm of model-based reinforcement learning (MBRL), such models play a pivotal role by enabling policy training through learning in imagination [1, 2, 3, 4], facilitating planning with look-ahead search [5, 6], or combining both approaches [7, 8]. While MBRL has achieved significant success in single-agent settings, extending these methodologies to multi-agent scenarios presents unique challenges, necessitating new approaches for multi-agent world modeling.

In multi-agent settings, where multiple agents simultaneously interact within a shared environment, two primary challenges emerge. First, the joint action space grows exponentially with the number of

---

\*Work done during the internship at TeleAI.
†Corresponding Author.

39th Conference on Neural Information Processing Systems (NeurIPS 2025).

agents [9, 10], making it computationally expensive to directly handle joint dynamics. Second, the complex interdependencies among agents [11] make it difficult to accurately capture how individual actions impact global state transitions. Current multi-agent world modeling approaches face a fundamental tradeoff. On one end of the spectrum, centralized modeling schemes directly capture full joint dynamics but incur computational costs that scale exponentially with the number of agents. On the other end, decentralized approaches [12, 13, 14] model individual agent dynamics separately and rely on additional mechanisms, such as sophisticated communication or aggregation modules, to recover the global state. However, this misalignment between decentralized model structure and the global Markov decision process (MDP) can impose inherent limitations on model accuracy and those communication or aggregation modules do not have explicit signal for supervision, further hindering the training. This tradeoff motivates a fundamental rethinking of the world model structure: Can we develop a centralized modeling scheme that maintains global consistency without auxiliary components in decentralized methods, while keeping computational complexity manageable as the number of agents increases?

To address this challenge, we adopt a sequential agent modeling perspective that processes agents' actions incrementally, as illustrated in Figure 1. Specifically, consider a multi-agent system at timestep $t$ with global state $s_t$. When all agents' actions $a_t^{1:n}$ are unknown, the next state $s_{t+1}$ remains highly uncertain. As agents' actions are progressively revealed, this uncertainty gradually decreases. This sequential uncertainty reduction process bears striking similarity to the reverse process in diffusion models [15, 16, 17], where generation is framed as iterative denoising from noise to clean samples.

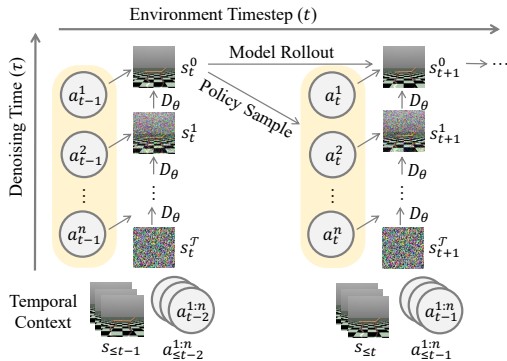

Inspired by this conceptual similarity and the recent success of diffusion models in image-based world modeling [18, 19, 20], we propose **D**iffusion-**I**nspired **M**ulti-**A**gent world model (DIMA), which reformulates multi-agent dynamics prediction as a modified conditional denoising process. Despite employing a centralized modeling scheme, DIMA achieves computational complexity that scales linearly with the state space dimensionality, regardless of the number of agents. We summarize our contributions as follows:

Figure 1: Illustration of the DIMA world model. From the temporal perspective, each environmental timestep is modeled as a complete denoising process, analogous to diffusion models. Within each timestep, we further consider an agent-wise perspective, where the introduction of each individual agent's action information represents a single denoising step, progressively reducing uncertainty about the next state.

- We leverage the connection between sequential agent modeling and diffusion processes to reformulate multi-agent dynamics prediction as a conditional denoising process. This enables a centralized modeling scheme that reduces complexity without additional communication mechanisms.
- We propose DIMA, a centralized multi-agent world model tailored for model-based MARL, and derive its corresponding evidence lower bound (ELBO), providing theoretical insights. We then instantiate DIMA within the EDM training framework [21] and integrate it into the learning-in-imagination paradigm for policy optimization.
- We evaluate DIMA on challenging continuous MARL benchmarks, including MAMuJoCo [22] and Bi-DexHands [23], in low-data regimes. Experimental results show that DIMA consistently improves the prediction accuracy of environment dynamics and outperforms both model-free and strong model-based MARL baselines in terms of sample efficiency and overall performance.

## 2 Preliminaries

### 2.1 Multi-Agent Systems as Dec-POMDP

We focus on fully cooperative multi-agent systems where all agents share a team reward signal. We formulate the system as a decentralized partially observable Markov decision process (Dec-POMDP) [11], which can be described by a tuple $(\mathcal{N}, \mathcal{S}, \mathcal{A}, P, R, \boldsymbol{\Omega}, \mathcal{O}, \gamma)$. $\mathcal{N} = \{1, ..., n\}$ denotes a set

of agents, $\mathcal{S}$ is the finite global state space, $\mathcal{A} = \prod_{i=1}^n \mathcal{A}^i$ is the product of finite action spaces of all agents, i.e., the joint action space, $P : \mathcal{S} \times \mathcal{A} \times \mathcal{S} \to [0, 1]$ is the global transition probability function, $R : \mathcal{S} \times \mathcal{A} \to \mathbb{R}$ is the shared reward function, $\mathbf{\Omega} = \prod_{i=1}^n \Omega^i$ is the product of finite observation spaces of all agents, i.e., the joint observation space, $\mathcal{O} = \{\mathcal{O}^i, i \in \mathcal{N}\}$ is the set of observing functions of all agents. $\mathcal{O}^i : \mathcal{S} \to \Omega^i$ maps global states to the observations for agent $i$, and $\gamma$ is the discount factor. Given a global state $s_t$ at timestep $t$, agent $i$ is restricted to obtaining solely its local observation $o_t^i = \mathcal{O}^i(s_t)$, takes an action $a_t^i$ drawn from its policy $\pi^i(\cdot|o_{\leq t}^i)$ based on the history of its local observations $o_{\leq t}^i$, which together with other agents' actions gives a joint action $\boldsymbol{a}_t = (a_t^1, ..., a_t^n) \in \mathcal{A}$, equivalently drawn from a joint policy $\boldsymbol{\pi}(\cdot|\boldsymbol{o}_{\leq t}) = \prod_{i=1}^n \pi^i(\cdot|o_{\leq t}^i)$. Then the agents receive a shared reward $r_t = R(s_t, \boldsymbol{a}_t)$, and the environment moves to next state $s_{t+1}$ with probability $P(s_{t+1}|s_t, \boldsymbol{a}_t)$. The aim of all agents is to learn a joint policy $\boldsymbol{\pi}$ that maximizes the expected discounted return $J(\boldsymbol{\pi}) = \mathbb{E}_{s_0, \boldsymbol{a}_0, ... \sim \boldsymbol{\pi}} \left[ \sum_{t=0}^\infty \gamma^t R(s_t, \boldsymbol{a}_t) \right]$. Note that recent approaches [12, 13, 14] build the multi-agent world models via modeling $P(\boldsymbol{o}_{t+1}|\boldsymbol{o}_t, \boldsymbol{a}_t)$ in the joint observation and action space, which mismatches with the transition formulation in Dec-POMDP. However, DIMA is trained to recover the well-defined global state transition $P(s_{t+1}|s_t, \boldsymbol{a}_t)$ according to the proposed multi-agent dynamics formulation.

## 2.2  Score-based Diffusion Models

In this work, we directly utilize the unified framework and the accompanying practical design choice of diffusion models introduced by Karras et al. [21].

**Notation.** Let us consider a diffusion process $\{\mathbf{x}^\tau\}_{\tau \in [0, \mathcal{T}]}$ indexed by a continuous time variable $\tau \in [0, \mathcal{T}]$, with corresponding marginals $\{p^\tau\}_{\tau \in [0, \mathcal{T}]}$, and boundary conditions $p^0 = p_{\text{data}}$ and $p^\mathcal{T} = p_{\text{prior}}$, where $p_{\text{prior}}$ is usually a pure Gaussian distribution in practical implementation. For clarity, we use the superscript $\tau$ to denote the diffusion process timestep and the subscript $t$ to denote the trajectory timestep.

**ODE Expression.** Song et al. [17] models the forward and reverse diffusion processes with stochastic differential equations (SDEs) which describe how the desired distribution of sample $\mathbf{x}$ evolves over time $\tau$. Assuming the stochasticity only comes from the initial sample $\mathbf{x}^\mathcal{T}$ of prior distribution $p_{\text{prior}}$, Karras et al. [21] expresses diffusion models via its corresponding probability flow ordinary differential equation (ODE) [17] which continuously increases or reduces the noise level of the image when moving forward or backward in time, respectively. The defining characteristic of the probability flow ODE is that evolving a sample $\mathbf{x}^{\tau_a} \sim p^{\tau_a}(\mathbf{x}) = p(\mathbf{x}; \sigma(\tau_a))$ from time $\tau_a$ to $\tau_b$ (either forward or backward in time) yields a sample $\mathbf{x}^{\tau_b} \sim p^{\tau_b}(\mathbf{x}) = p(\mathbf{x}; \sigma(\tau_b))$, where $\sigma(\tau)$ is a schedule that defines the desired noise level at time $\tau$. It is described by

$$d\mathbf{x} = -\dot{\sigma}(\tau)\sigma(\tau)\nabla_{\mathbf{x}} \log p^\tau(\mathbf{x}) \, d\tau,$$

where the dot denotes a time derivative. $\nabla_{\mathbf{x}} \log p^\tau(\mathbf{x})$ is the score function [24] associated with the marginals $\{p^\tau\}_{\tau \in [0, \mathcal{T}]}$ along the process. Equipped with the score function, we can thus smoothly mold random noise into data for sample generation, or diffuse a data point into random noise.

**Denoising Score Matching.** By using the score matching objective [24], we can evaluate the score function easily. Specifically, $D_\theta(\mathbf{x}; \tau)$ is a parameterized denoiser function that minimizes the expected $L_2$ denoising error for samples $\mathbf{x}^0$ drawn from $p_{\text{data}}$ for every $\sigma(\tau)$,

$$\mathcal{L}(\theta) = \mathbb{E}_{\mathbf{x}^0 \sim p_{\text{data}}(\mathbf{x}), \mathbf{x}^\tau \sim p(\mathbf{x}^\tau|\mathbf{x}^0)}[\|D_\theta(\mathbf{x}^\tau; \tau) - \mathbf{x}^0\|^2], \tag{1}$$

where $\mathbf{x}^\tau \sim p(\mathbf{x}^\tau|\mathbf{x}^0)$ denotes that $\mathbf{x}^\tau$ is obtained by applying Gaussian noise of scale $\sigma(\tau)$ to clean sample $\mathbf{x}^0$. Then the score estimation can be given by $\nabla_{\mathbf{x}} \log p^\tau(\mathbf{x}) = (D_\theta(\mathbf{x}; \tau) - \mathbf{x})/\sigma(\tau)^2$ at any given time $\tau$. Thanks to this estimation, we can solve the ODE by numerical integration, i.e., taking finite steps over discrete time intervals with the help of various ODE solvers.

## 3  Methodology

In the following, we first elucidate our proposed formulation of modeling multi-agent dynamics from a diffusion-inspired perspective in §3.1. Based on this formulation, we derive the corresponding ELBO and score matching objective for implementing the diffusion model that incorporates such a perspective. Then, we describe the behavior learning process within the world model in §3.2.

### 3.1 Modeling Multi-Agent Dynamics from a Diffusion-Inspired Perspective

Given a dataset $\{(\boldsymbol{o}_1, s_1, \boldsymbol{a}_1, r_1, \ldots, \boldsymbol{o}_{T_i}, s_{T_i}, \boldsymbol{a}_{T_i}, r_{T_i})\}_i$ containing all collected episodes, the aim of the multi-agent world model is to precisely predict how the next state is like based on an action intervention, i.e., recovering the unknown ground truth environment dynamics $P(s_{t+1}|s_t, \boldsymbol{a}_t)$.

**Diffusion-Inspired Formulation.** Supposing there are $n$ agents $\{1, 2, \ldots, n\}$ and $n$ noise levels $\{\sigma_n, \ldots, \sigma_2, \sigma_1\}$ that satisfy $\sigma_{\max} = \sigma_n > \cdots > \sigma_1 > \sigma_0 = 0$, the noisy sample $s_{t+1}^{(i)}$ is corrupted from the clean next state $s_{t+1}^{(0)} := s_{t+1}$ by adding noise of the corresponding level $\sigma_i$. Note that here we use the superscript to denote the diffusion process timestep except for the action notation $a$. Following the definition in [25], we start by defining a similar conditional Markovian forward diffusion process $\hat{q}$,

$$\hat{q}(s_{t+1}^{(0)}) := p(s_{t+1}), \tag{2}$$

$$\hat{q}(s_{t+1}^{(k+1)}|s_{t+1}^{(k)}, s_t, a_t^{1:n}) := q(s_{t+1}^{(k+1)}|s_{t+1}^{(k)}), \tag{3}$$

$$\hat{q}(s_{t+1}^{(1):(n)}|s_{t+1}^{(0)}, s_t, a_t^{1:n}) := \prod_{k=1}^{n} \hat{q}(s_{t+1}^{(k+1)}|s_{t+1}^{(k)}, s_t, a_t^{1:n}), \tag{4}$$

where $q$ denotes the unconditional forward diffusion process. While the conditional forward diffusion process $\hat{q}$ is conditioned on the control signal $(s_t, a_t^{1:n})$, we can prove that it behaves exactly like the unconditional one $q$. The following equations hold,

$$\hat{q}(s_{t+1}^{(k+1)}|s_{t+1}^{(k)}) = \hat{q}(s_{t+1}^{(k+1)}|s_{t+1}^{(k)}, s_t, a_t^{1:n}), \ \hat{q}(s_{t+1}^{(1):(n)}|s_{t+1}^{(0)}) = q(s_{t+1}^{(1):(n)}|s_{t+1}^{(0)}). \tag{5}$$

The detailed proof is referred to Dhariwal and Nichol [25]. Since the above equations suggest that the forward diffusion process is independent of the control signal $(s_t, a_t^{1:n})$, we can now fully focus on describing our formulation via the conditional reverse diffusion process.

To describe how the predicted next state gets sharpened progressively with sequentially given action of each agent, we have to specify the conditioning order. Without loss of generality, we adopt the descending order of agent id $(n, n-1, \ldots, 1)$ as the conditioning order. Formally, we make the following assumption in terms of the global state transition.

**Assumption 1** (Diffusion-Inspired Decomposition of Multi-Agent Dynamics). *In our diffusion-inspired formulation with the descending order of agent id $(n, n-1, \ldots, 1)$ as the conditioning order, the global state transition $P(s_{t+1}|s_t, a_t^{1:n})$ yields the next state in a manner akin to a typical reverse diffusion process, i.e., satisfying*

$$P(s_{t+1}, s_{t+1}^{(1):(n)}|s_t, a_t^{1:n}) = p(s_{t+1}^{(n)}) \prod_{k=1}^{n} p(s_{t+1}^{(k-1)}|s_{t+1}^{(k)}, a_t^k, s_t), \tag{6}$$

*where $s_{t+1}^{(n)}$ is corrupted with the noise of maximum level $\sigma_n$, practically indistinguishable from pure Gaussian noise.*

Under the assumption, we have the following new form of Evidence Lower Bound (ELBO) on the $\log P(s_{t+1}|s_t, a_t^{1:n})$.

**Theorem 2** (ELBO under the Diffusion-Inspired Formulation). *Under Assumption 1, the log-likelihood of the multi-agent global state transition (i.e., the evidence of the transition) is lower bounded as follows,*

$$\log P(s_{t+1}|s_t, a_t^{1:n}) \geq \underbrace{\mathbb{E}_{q(s_{t+1}^{(1)}|s_{t+1}^{(0)})}[\log p(s_{t+1}^{(0)}|s_{t+1}^{(1)}, a_t^1, s_t)]}_{\text{reconstruction term}} - \underbrace{D_{\mathrm{KL}}(q(s_{t+1}^{(n)}|s_{t+1}^{(0)})\|p(s_{t+1}^{(n)}))}_{\text{prior matching term}}$$

$$- \sum_{k=2}^{n} \underbrace{\mathbb{E}_{q(s_{t+1}^{(k)}|s_{t+1}^{(0)})}\left[D_{\mathrm{KL}}(q(s_{t+1}^{(k-1)}|s_{t+1}^{(k)}, s_{t+1}^{(0)})\|p(s_{t+1}^{(k-1)}|s_{t+1}^{(k)}, a_t^k, s_t))\right]}_{\text{denoising matching term}}. \tag{7}$$

The detailed proof is deferred to §A. The *denoising matching term* in Eq. (7) secretly reveals that we can learn a parameterized denoising intermediate step $p_\theta(s_{t+1}^{(k-1)}|s_{t+1}^{(k)}, a_t^k, s_t)$ that matches

the tractable ground-truth denoising intermediate step $q(s_{t+1}^{(k-1)}|s_{t+1}^{(k)}, s_{t+1}^{(0)})$, thereby realizing the formulation we propose. When we utilize Gaussian noise for corruption in the forward diffusion process, the *denoising matching term* can be simplified as a variant of Eq. (1):

$$\mathcal{L}(\theta) = \mathbb{E}\left[\sum_{k=1}^{n}\|D_\theta(s_{t+1}^{(k)};\sigma_k, s_t, a_t^k) - s_{t+1}\|^2\right], \text{ given the order } (n, \ldots, 2, 1) \qquad (8)$$

However, there are still two properties to be incorporated into Eq. (8). **(i) Permutation Invariance.** Note that our formulation merely provides a novel perspective for modeling multi-agent dynamics, rather than changing the underlying mechanism of global state transitions. In other words, regardless of how the conditioning order of $a_t^{1:n}$ is specified, the next state should remain unchanged given the same current state and joint action, i.e., exhibiting *permutation invariance*. Therefore, for any possible order $\rho = (i_1, i_2, \ldots, i_n)$ uniformly sampled from the whole permutation set Perm$\{1, 2, \ldots, n\}$, we should optimize an expectation of Eq. (8) over the whole permutation set. **(ii) Condition-Independent Noising Process.** According to Eqs. (2)-(5), the conditional forward diffusion process is independent of the conditions. It allows us to randomly sample the noise levels $\{\sigma_1, \ldots, \sigma_n\}$ with the predefined continuous-time noise scheduler $\sigma(\tau)$ in §2.2.

Putting the above two together, we finally derive the optimization objective of DIMA,

$$\mathcal{L}(\theta) = \mathbb{E}_{\{\sigma_1,\ldots,\sigma_n\}\sim\sigma(\tau)}\mathbb{E}_{\rho\sim\text{Perm}\{1,2,\ldots,n\}}\left[\sum_{k=1}^{n}\|D_\theta(s_{t+1}^{(k)};\sigma_k, s_t, a_t^{i_k}) - s_{t+1}\|^2\right]$$

$$= \mathbb{E}_\tau\mathbb{E}_{k\sim\text{Uniform}\{1,2,\ldots,n\}}\left[\|D_\theta(s_{t+1}^\tau;\sigma(\tau), s_t, a_t^k) - s_{t+1}\|^2\right], \qquad (9)$$

where $k \sim \text{Uniform}\{1, 2, \ldots, n\}$ indicates that the agent index $k$ is uniformly sampled from the set $\{1, 2, \ldots, n\}$.

**Comparison with Conventional Approaches.** We present a concise illustration to highlight the fundamental difference between our DIMA and recent diffusion-based methods [26, 27] in modeling multi-agent dynamics. As shown in Figure 2, recent methods attempt to inject the entire joint action information into the progressively denoised next state at every intermediate step, whereas DIMA incorporates only a single agent's action at each step. Denoting the state space size as $|\mathcal{S}|$ and the individual action space size as $|\mathcal{A}|$, DIMA compresses the relevant information from a $|\mathcal{S}| \times |\mathcal{A}| \times |\mathcal{S}|$ space into a $|\mathcal{S}|$ space for each intermediate state transition $p_\theta(s_{t+1}^{(k-1)}|s_{t+1}^{(k)}, a_t^k, s_t)$. In contrast, existing methods must handle a significantly higher cost due to compressing information from a

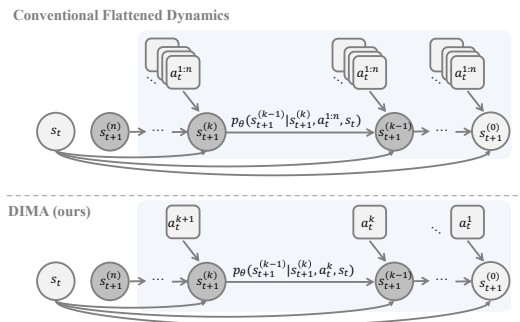

Figure 2: Comparison between conventional flattened multi-agent modeling and DIMA's sequential agent modeling. Light gray indicates clean states; dark gray indicates noisy states.

$|\mathcal{S}| \times |\mathcal{A}|^n \times |\mathcal{S}|$ space into $|\mathcal{S}|$. This simple qualitative analysis demonstrates that despite modeling multi-agent dynamics in a centralized manner, DIMA enjoys a linear complexity in modeling difficulty with respect to the number of agents.

**Practical Implementation.** Inspired by the success of DIAMOND [18], a powerful single-agent diffusion-based world model, we adopt a similar design choice and employ the EDM framework [21] to effectively train the desired diffusion model. Specifically, the denoiser $D_\theta$ is reparameterized using the EDM preconditioners as follows:

$$D_\theta(s_{t+1}^\tau;\sigma(\tau), s_t, a_t^k) = c_{\text{skip}}^\tau s_{t+1}^\tau + c_{\text{out}}^\tau F_\theta(c_{\text{in}}^\tau s_{t+1}^\tau; c_{\text{noise}}^\tau, s_t, a_t^k), \qquad (10)$$

where $F_\theta$ is the neural network. These preconditioners $(c_{\text{skip}}^\tau, c_{\text{out}}^\tau, c_{\text{in}}^\tau, c_{\text{noise}}^\tau)$ are detailed in §B. In addition, we incorporate two practical techniques to further improve DIMA's predictive performance: (i) we maintain a running mean and standard deviation of global states to normalize the state before training, ensuring stable dynamics ranges; (ii) we augment the model input with a fixed window of past $k$ global states and joint actions to provide richer temporal context for next-state prediction.

### 3.2 Learning Behaviors in Imagination

To support reinforcement learning with imagined rollouts, we pair DIMA with two necessary components. The first is a reward and termination model $f_\phi$ where reward prediction and termination pre-

diction are framed as scalar regression and binary classification tasks, respectively. Motivated by the advanced sequence modeling capability of Transformer [28], we employ a Transformer architecture as the backbone. As illustrated in Figure 3, the model takes sequences of $(\dots, s_t, a_t^{1:n}, s_{t+1}, a_{t+1}^{1:n}, \dots)$ as input and predicts reward and termination at each timestep via two separate 3-layer multilayer perceptron (MLP) heads on top of the shared output embedding. The model is built upon MinGPT implementation [29]. The second component is a special auto-encoder $g_\varphi(o_t^{1:n}|s_t)$ that encodes the global state $s_t$ into a compact latent space and decodes it into the joint observation $o_t^{1:n}$. We implement this using a simple yet effective VQ-VAE [30] with Finite Scalar Quantization [31]. We adopt an actor-critic framework to learn the behavior policy of each agent, where the actor and critic are parameterized by two 3-layer MLPs, $\pi_\psi(a_t^i|o_t^i)$ and $V_\xi(s_t)$, respectively.

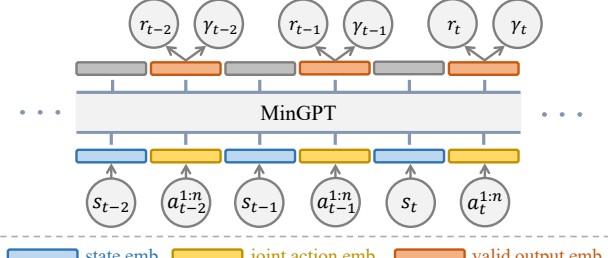

Figure 3: Overview of the reward and termination model. DIMA addresses reward and termination prediction from a global perspective using a transformer architecture to capture temporal correlations. Both functions share the same backbone with separate prediction heads.

Thanks to DIMA's global state transition predictions, we can leverage oracle information from the global state to train a centralized critic, which in turn guides the optimization of decentralized agent actors. This naturally aligns with the centralized training with decentralized execution (CTDE) paradigm commonly used in model-free MARL [32, 33, 34]. Moreover, this provides a clear advantage over recent model-based MARL methods that also rely on learning in imagination [12, 14]. As these methods typically model local observation dynamics for scalability, they lose the benefits of accessing oracle global state, which our approach fully exploits. Here we train the actor and critic with MAPPO [34]. $\lambda$-return [1] is used as the target to update the value function. The details of behavior learning objectives and algorithmic description are presented in §C and §F, respectively.

As we evaluate DIMA under the learning in imagination paradigm, our approach iteratively executes a cycle that comprises three steps: (i) collecting experience by executing the policy, (ii) updating the world model with the collected experience, and (iii) learning the policy through imagined rollouts within the learned world model. Note that throughout the whole procedure, the historical experiences stored in the replay buffer are only used for training the world model, while the policy is optimized through unlimited imagined trajectories generated by the world model.

## 4    Related Works

**Diffusion Model for RL.**  Diffusion models [16, 35] have been applied in reinforcement learning (RL) for its strong generation capability. Specifically, they are capable of modeling complex action distributions in online RL [36, 37, 38, 39], offline policy learning [40, 41, 42], and imitation learning [43, 44, 45]. Other works also adopt diffusion models as planners to generate state-action sequences [46, 47, 48]. Recently, diffusion policies are also used as an action expert to combine with LLMs and obtain visual-language-action model [49, 50]. For dynamics modeling, diffusion models have been employed as alternatives to autoregressive models to learn the complex transition function of MDPs [18, 51, 19], while they are limited in the single-agent domain. MADiff [26] learns the distribution of the whole trajectory in the offline multi-agent settings, without modeling the step-wise transitions. Other works [52, 53, 54] treat diffusion-based world modeling as a video generation problem without taking actions as a condition, limiting their abilities. In contrast, our proposed DIMA predicts future states based on sequential action conditions, which effectively builds the multi-agent world model.

**Multi-Agent RL.** In cooperative MARL, agents coordinate to maximize a joint reward function. Centralized Training with Decentralized Execution (CTDE) [55] is a foundational framework that leverages the global state of agents during training to facilitate policy learning while relying on partial information during execution. CTDE framework serves the basis for both value-based [32, 33, 56] and policy-based MARL methods [57, 34, 58]. Additionally, some works reformulate MARL as a sequential decision-making problem [59, 60, 61], offering insight into sequential denoising in diffusion-based dynamics. Model-based MARL has gained significant attention for its ability to explicitly model the underlying MDPs in multi-agent environments. Notable examples include

MAZero [13], which adapts MuZero-style planning with MCTS, and Dreamer-based methods [12, 62, 14], which leverage learning in imaginations for multi-agent setups [1, 2, 63]. These approaches have demonstrated the potential of model-based methods to improve coordination in multi-agent systems.

More recently, diffusion models have been introduced into MARL to enhance coordination and trajectory modeling, motivated by their advanced modeling capabilities. MADiff [26] first introduces diffusion models in MARL through offline trajectory learning via attention-based diffusion. Subsequent works have extended the use of diffusion models in MARL. Specifically, DoF [64] investigates offline MARL by factorizing a centralized diffusion model into multiple sub-models, aligning with the CTDE framework. Similarly, MADiTS [27] explores diffusion-based data augmentation by stitching high-quality coordination segments together. While effective, these methods primarily use diffusion models as goal-conditioned trajectory generators, failing to account for the underlying multi-agent dynamics. Our proposed DIMA addresses this research gap by constructing an effective world model that explicitly captures the multi-agent dynamics. By leveraging the strengths of diffusion-inspired modeling, DIMA assists policy training and improves the overall performance of MARL.

## 5 Experiments

### 5.1 Experiments Setup

**Environments.** We evaluate our method on two widely-used multi-agent continuous control benchmarks requiring heterogeneous-agent cooperation: Multi-Agent MuJoCo (MAMuJoCo) [22] and Bimanual Dexterous Hands (Bi-DexHands) [23]. MAMuJoCo extends MuJoCo [65] to multi-agent settings by partitioning a robot into agents controlling different degrees of freedom (DoFs), requiring coordination for coherent movement. We use seven agent-partitioning settings: HalfCheetah [2x3, 3x2, and 6x1]; Walker [2x3 and 3x2]; and Ant [2x4 and 4x2]. Bi-DexHands features dual ShadowRobot hands (26 DoFs each) performing precise bimanual manipulation. We evaluate on four tasks: *ShadowHandPen*, *ShadowHandDoorOpenOutward*, *ShadowHandDoorOpenInward*, and *ShadowHandBottleCap*. To highlight the sample efficiency of learning in imaginations, we adopt a low-data regime [66], limiting real-environment samples to 1M for MAMuJoCo and 300k for Bi-DexHands, adjusted for their different episode lengths. In model-based MARL where policies are learned in imaginations, performance directly reflects the accuracy of the world model, enabling transparent evaluation.

**Baselines.** We compare DIMA against two strong model-based baselines with the same policy learning paradigm as ours – MAMBA [12] and MARIE [14]. MAMBA extends DreamerV2 [39] to the multi-agent context and establishes an effective Recurrent State Space Model (RSSM)-based world model. MARIE incorporates Transformer-based autoregressive world modeling [3] with CTDE principle and demonstrates remarkable sample efficiency on the benchmark with discrete action space. We also compare DIMA with strong model-free baselines, including two on-policy algorithms MAPPO [34] and HAPPO [58], and an off-policy algorithm HASAC [67, 68]. HASAC is a heterogeneous-agent extension of SAC [69] which is well known for its high sample efficiency. Each algorithm is evaluated using 4 random seeds per scenario. For each random seed, we report the averaged episode return across 10 evaluation episodes at fixed intervals of environment steps. To ensure a fair comparison, we restrict the imagination horizon $H = 15$ for all model-based algorithms. Results of MARIE would not be reported in Bi-DexHands due to severe out-of-memory issues under our available computational resources.

### 5.2 Main Results

**DIMA consistently outperforms all evaluated baselines across a wide range of multi-agent continuous control tasks, achieving superior sample efficiency and higher final returns.** As shown in Figure 4 and 5, DIMA exhibits rapid and consistent policy convergence across all chosen MAMuJoCo and Bi-DexHands tasks, while other model-based baselines fail to demonstrate such stable learning behavior. This highlights the advantage of our approach in leveraging an effective world modeling formulation that is better aligned with the global state transitions of the environment. MARIE and MAMBA suffer from a mismatch with the true global transition dynamics inherent in Dec-POMDPs due to their integration of local dynamics modeling with the CTDE principle. This discrepancy potentially imposes an inherent limitation on model accuracy, particularly in environments like MAMuJoCo where inter-agent dependencies are strongly correlated. Although

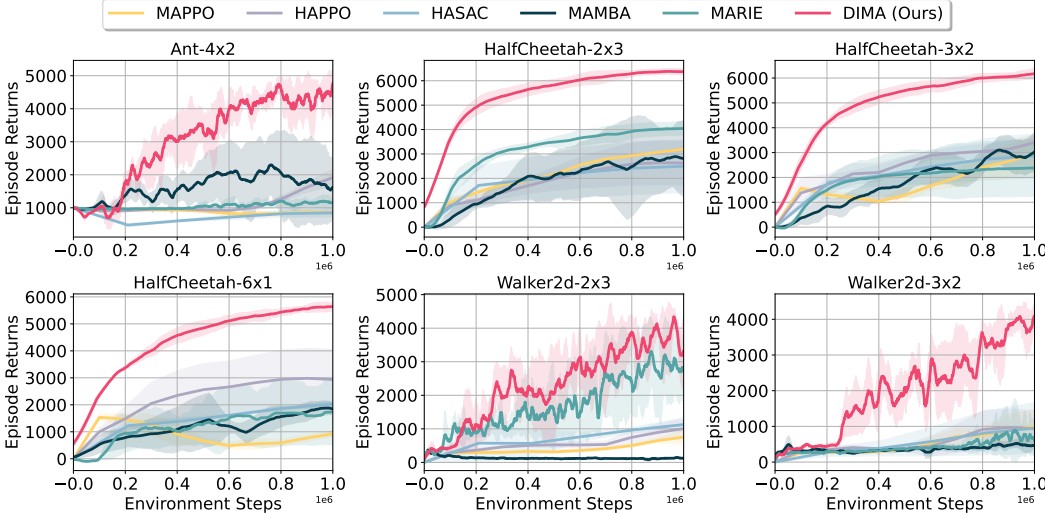

Figure 4: Curves of averaged episode returns for all methods in MAMuJoCo.

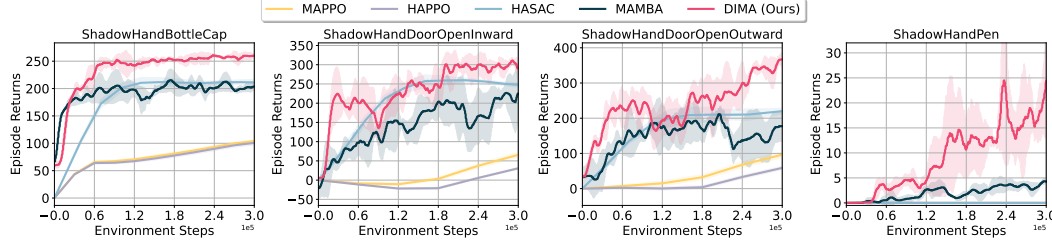

Figure 5: Curves of averaged episode returns for all methods in Bi-DexHands.

enjoying world modeling complexity at a linear rate, such architectural misalignment limits their scalability to highly coupled settings. Interestingly, MARIE and MAMBA perform comparably to—or even worse than sample-inefficient on-policy model-free methods HAPPO and MAPPO (e.g., in HalfCheetah [3x2, 6x1]), whereas DIMA consistently demonstrates superior performance. This performance gain reflects the accuracy and robustness of DIMA in enabling more precise and reliable imaginations for policy optimization.

A similar trend is observed in the Bi-DexHands benchmark, characterized by the control of two dexterous hands, each with 26 DoFs (i.e., $a_t^i \in \mathbb{R}^{26}$). Benefiting from the expressiveness of the diffusion model, DIMA is able to more accurately capture especially sophisticated and contact-rich dynamics. By learning a denoising generative process under our formulation, DIMA enables more faithful representations of the underlying transition distribution and leads to more stable, coherent imagined trajectories for downstream policy learning, compared to RSSM-based models. As a result, DIMA substantially improves the learning of dexterous manipulation policies in scenarios requiring fine-grained, high-precision coordination. The numerical results are further provided in §E.

## 5.3 Model Analysis

**DIMA demonstrates substantially more accurate and stable long-horizon predictions than existing multi-agent world models.** To better evaluate the model capabilities among MAMBA, MARIE and our DIMA, we visualize their imagined trajectories alongside the ground truth (GT) on Ant [2x4] task. As visualized in Figure 6, DIMA generates a consistent imagined trajectory that closely aligns with the ground truth (GT) across the full prediction horizon $H = 15$, maintaining coherent agent structures and motion patterns. In contrast, MARIE and MAMBA both exhibit significant degradation as the horizon extends, and suffer from varying degrees of distortions. These issues are especially pronounced at challenging future timesteps such as $t = 4$ and $t = 12$, highlighted by red bounding boxes. The qualitative results underscore DIMA's superior modeling capability and stability in capturing complex continuous multi-agent control dynamics, which is critical for generating reliable imagined rollouts that support sample-efficient policy learning.

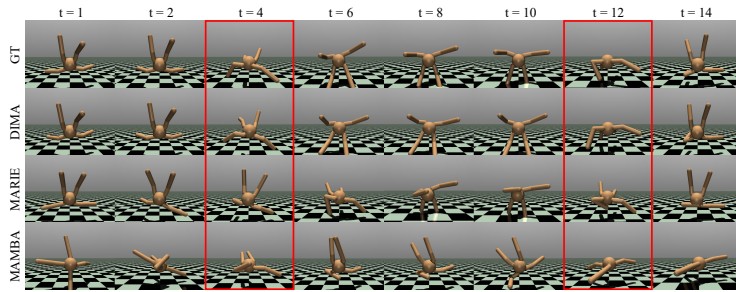

Figure 6: Reconstructions of long-term predictions of different multi-agent world models. We qualitatively compare the reconstruction quality of different multi-agent world models. Each model perform forward imagination over a horizon of $H = 15$.

**DIMA scales robustly to longer horizons with significantly lower compounding error.** DIMA demonstrates superior scalability and robustly mitigates compounding errors, a capability we validated by testing on prediction horizons ($H = 25$) **significantly longer than those seen during training** ($H = 15$). This setup provides a rigorous out-of-distribution (OOD) generalization test for auto-regressive world models. The results in Table 1 are conclusive: on the Ant [2x4] benchmark, DIMA not only exhibits the lowest L1 accumulated observation and reward errors at the training horizon ($H = 15$) **but significantly widens this performance gap** against MARIE and MAMBA at the unseen $H = 25$ horizon. This finding underscores DIMA's effectiveness in long-range predictive accuracy. Downstream policy performance using this extended imagination horizon is detailed in §E.2.

Table 1: Accumulated observation and reward errors at extended prediction horizons ($H = 25$) on Ant [2x4]. Results are averaged over 100 trajectory segments. DIMA exhibits the lowest compounding error.

| Methods | Obs Accumulation Errors | | Rew Accumulation Errors | |
|---|---|---|---|---|
| | @ $H = 15$ | @ $H = 25$ | @ $H = 15$ | @ $H = 25$ |
| DIMA | $\mathbf{2.42}_{\pm 0.93}$ | $\mathbf{4.32}_{\pm 1.44}$ | $\mathbf{15.01}_{\pm 9.88}$ | $\mathbf{31.07}_{\pm 17.63}$ |
| MARIE | $3.54_{\pm 1.60}$ | $6.62_{\pm 2.68}$ | $21.64_{\pm 17.51}$ | $47.66_{\pm 32.02}$ |
| MAMBA | $3.98_{\pm 1.54}$ | $6.85_{\pm 2.51}$ | $38.90_{\pm 24.48}$ | $71.67_{\pm 41.78}$ |

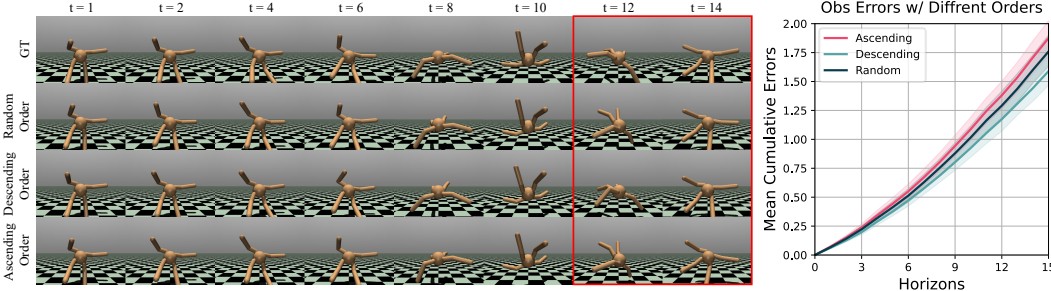

Figure 7: Visualization of long-term predictions with different conditioning orders, together with the accompanying cumulative observation errors curve.

**DIMA effectively preserves permutation invariance over long-horizon multi-agent predictions.** To validate whether and how DIMA exhibits the desired *permutation invariance* property elaborated in §3.1, we evaluate its unrolling behavior under different conditioning orders of agent actions. As shown in Figure 7 (left), we generate imagined rollouts from the same initial state and joint action set, but vary the conditioning order using three representative orders: random, ascending and descending w.r.t. agent ids. DIMA produces visually consistent rollouts across different orders until notable visual differences emerges at $t = 12$, highlighted by the red bounding box. This demonstrates that DIMA maintains this consistency effectively up to a long prediction horizon at least $H = 10$. To further quantify the consistency, we plot the mean cumulative observation errors over prediction horizons under each order, as depicted in Figure 7 (right). The resulting curves seems quite aligned, with no significant deviation among the three conditions, indicating that DIMA exhibits the *permutation invariance* property within a considerably long horizon via optimizing Eq. (9). Details of this experiment setup are provided in §E.

## 5.4 Ablation Study

**Our proposed formulation improves sample efficiency and stability in low-data regimes.** To evaluate the core contribution of DIMA's agent-wise sequential modeling, we compare it against a "Joint" modeling baseline, which adopts the conventional centralized approach of conditioning on the full joint action $a_t^{1:n}$ at every denoising step. The primary benefit of our sequential formulation is the reduction of modeling complexity from an exponential to a linear dependency on the number of agents. This reduction is particularly impactful in low-data regimes, where a simpler model can learn more effectively. We conducted experiments on several Bi-DexHands tasks with 8 independent runs. As shown in Table 2, on the *ShadowHandBottleCap* task, our sequential approach (DIMA) achieves higher returns and lower variance at 100k and 150k steps. As the data budget increases (200k-300k), the performance of the joint model catches up, and both methods converge to a similar performance. This aligns with our hypothesis: sequential modeling provides a significant sample efficiency boost when data is scarce. This benefit is even more pronounced in more complex tasks. Table 3 shows that on *DoorOpenOutward* and *DoorOpenInward*, DIMA (Sequential) maintains a clear performance advantage and reduced variance over the Joint model even at the 300k step limit. This demonstrates that for harder tasks, the reduced modeling complexity of sequential modeling remains beneficial for longer, leading to more stable and effective policy learning. This addresses the key request from the review process to validate the benefit of sequential modeling.

Table 2: Ablation study on **ShadowHandBottleCap** comparing sequential (DIMA) vs. joint modeling under varying data budgets (8 runs). Sequential modeling shows superior performance and lower variance in lower-data regimes.

| Method | 100K Steps | 150K Steps | 200K Steps | 250K Steps | 300K Steps |
|---|---|---|---|---|---|
| Joint | $234.1_{\pm 20.6}$ | $238.6_{\pm 22.9}$ | $246.7_{\pm 10.9}$ | $243.7_{\pm 18.2}$ | $255.2_{\pm 7.0}$ |
| Sequential (Ours) | $\mathbf{251.8}_{\pm 17.3}$ | $\mathbf{248.2}_{\pm 11.6}$ | $246.3_{\pm 14.6}$ | $\mathbf{251.9}_{\pm 12.7}$ | $249.2_{\pm 10.7}$ |

Table 3: Ablation study on complex Bi-DexHands tasks at 300k steps (8 runs). The advantage of sequential modeling persists in more challenging environments.

| Method | DoorOpenOutward @ 300K steps | DoorOpenInward @ 300K steps |
|---|---|---|
| Joint | $302.5_{\pm 76.9}$ | $235.1_{\pm 68.1}$ |
| Sequential (Ours) | $\mathbf{352.4}_{\pm 40.5}$ | $\mathbf{290.3}_{\pm 30.4}$ |

**Sequential Modeling Retains Full Predictive Accuracy with Reduced Complexity.** We conducted a direct empirical comparison of *prediction error* between sequential and joint modeling, independent of downstream policy optimization, to validate our design choice. We utilized a 1M-step replay dataset from HASAC, training both models on the first 500k transitions and evaluating their accumulated L1 observation errors on a held-out set of the final 500k transitions (averaged over 100 segments). As shown in **Table 8 from §E.4**, our sequential model achieves predictive accuracy that is statistically on par with the more complex joint modeling approach. Across three challenging Bi-DexHands tasks and at both $H = 15$ and extended $H = 20$ horizons, the error metrics are statistically indistinguishable when considering the standard deviations. It provides direct, quantitative evidence that the reduced modeling complexity (and associated benefits, e.g., computational efficiency) of our sequential approach comes at no cost to raw predictive capability. This result confirms that the added complexity of joint modeling is unnecessary for achieving high-fidelity predictions in these environments.

## 6 Conclusion

This paper presented a multi-agent world model motivated by the conceptual similarity between the progressive denoising process and the incremental reduction of uncertainty in predicting the global next state in MARL. Then, we propose DIMA that models multi-agent dynamics from a centralized perspective while achieving reduced complexity, seamlessly aligning the world model with the underlying MDPs to obtain more accurate predictions. To validate the efficacy of DIMA, we integrated it into the learning-in-imagination training scheme and conducted extensive experiments on the MAMuJoCo and Bi-DexHands benchmarks. The results demonstrated DIMA's superior accuracy and robustness in predicting environment dynamics, as well as its ability to enhance sample efficiency

and overall performance. Despite its effectiveness, DIMA may encounter scalability challenges when applied to large-scale multi-agent systems with hundreds of agents. To address this, we plan to explore grouping techniques to further extend DIMA's applicability and scalability in future work.

## Acknowledgments

We would like to thank Liyuan Mao for his insightful discussions and comments, and Jiaqi Peng for his generous help. This work is supported by the National Natural Science Foundation of China (Grant No.62306242), the Young Elite Scientists Sponsorship Program by CAST (Grant No. 2024QNRC001), the Yangfan Project of the Shanghai (Grant No.23YF11462200), and the General Research Fund (GRF 16209124).

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

# A    Proof of Theorem 2

*Proof.* Given the log-likelihood of the global state transition, we have the following:

$$
\log P(s_{t+1}|s_t, a_t^{1:N}) = \log \int p(s_{t+1}, s_{t+1}^{(1):(n)}|s_t, a_t^{1:n})\, ds_{t+1}^{(1):(n)}
$$

$$
= \log \int \frac{p(s_{t+1}, s_{t+1}^{(1):(n)}|s_t, a_t^{(1):(n)})\hat{q}(s_{t+1}^{(1):(n)}|s_t, a_t^{1:n}, s_{t+1})}{\hat{q}(s_{t+1}^{(1):(n)}|s_t, a_t^{1:n}, s_{t+1})}\, ds_{t+1}^{(1):(n)}
$$

$$
= \log \mathbb{E}_{\hat{q}(s_{t+1}^{(1):(n)}|s_t, a_t^{1:n}, s_{t+1})} \left[ \frac{p(s_{t+1}, s_{t+1}^{(1):(n)}|s_t, a_t^{1:n})}{\hat{q}(s_{t+1}^{(1):(n)}|s_t, a_t^{1:n}, s_{t+1})} \right]
$$

$$
\geq \mathbb{E}_{\hat{q}(s_{t+1}^{(1):(n)}|s_t, a_t^{1:n}, s_{t+1})} \left[ \log \frac{p(s_{t+1}, s_{t+1}^{(1):(n)}|s_t, a_t^{1:n})}{\hat{q}(s_{t+1}^{(1):(n)}|s_t, a_t^{1:n}, s_{t+1})} \right], \tag{11}
$$

where the last inequality results from Jensen's inequality. Under the definition and property of the conditional Markovian forward diffusion process $\hat{q}$ in Eqs. (2)–(5) and Assumption 1, we can rewrite Eq. (11) as follows,

$$
\log P(s_{t+1}|s_t, a_t^{1:N}) \geq \mathbb{E}_{\hat{q}(s_{t+1}^{(1):(n)}|s_{t+1})} \left[ \log \frac{p(s_{t+1}^{(n)}) \prod_{k=1}^{n} p(s_{t+1}^{(k-1)}|s_{t+1}^{(k)}, s_t, a_t^k))}{\prod_{k=1}^{n} \hat{q}(s_{t+1}^{(k)}|s_{t+1}^{(k-1)})} \right], \tag{12}
$$

where we denote $s_{t+1} := s_{t+1}^{(0)}$. Then, RHS of Eq. (12) can be further simplified,

$$
\mathbb{E}_{\hat{q}(s_{t+1}^{(1):(n)}|s_{t+1})} \left[ \log \frac{p(s_{t+1}^{(n)}) \prod_{k=1}^{n} p(s_{t+1}^{(k-1)}|s_{t+1}^{(k)}, s_t, a_t^k))}{\prod_{k=1}^{n} \hat{q}(s_{t+1}^{(k)}|s_{t+1}^{(k-1)})} \right]
$$

$$
= \mathbb{E}_{\hat{q}(s_{t+1}^{(1):(n)}|s_{t+1})} \left[ \log \frac{p(s_{t+1}^{(n)})p(s_{t+1}^{(0)}|s_{t+1}^{(1)}, s_t, a_t^1)}{\hat{q}(s_{t+1}^{(1)}|s_{t+1}^{(0)})} + \log \prod_{k=2}^{n} \frac{p(s_{t+1}^{(k-1)}|s_{t+1}^{(k)}, s_t, a_t^k)}{\hat{q}(s_{t+1}^{(k)}|s_{t+1}^{(k-1)})} \right]
$$

$$
= \mathbb{E}_{\hat{q}(s_{t+1}^{(1):(n)}|s_{t+1})} \left[ \log \frac{p(s_{t+1}^{(n)})p(s_{t+1}^{(0)}|s_{t+1}^{(1)}, s_t, a_t^1)}{\hat{q}(s_{t+1}^{(1)}|s_{t+1}^{(0)})} + \log \prod_{k=2}^{n} \frac{p(s_{t+1}^{(k-1)}|s_{t+1}^{(k)}, s_t, a_t^k)}{\frac{\hat{q}(s_{t+1}^{(k-1)}|s_{t+1}^{(k)}, s_{t+1}^{(0)})\hat{q}(s_{t+1}^{(k)}|s_{t+1}^{(0)})}{q(s_{t+1}^{(k-1)}|s_{t+1}^{(0)})}} \right]
$$

$$
= \mathbb{E}_{\hat{q}(s_{t+1}^{(1):(n)}|s_{t+1})} \left[ \log \frac{p(s_{t+1}^{(n)})p(s_{t+1}^{(0)}|s_{t+1}^{(1)}, s_t, a_t^1)}{\hat{q}(s_{t+1}^{(1)}|s_{t+1}^{(0)})} + \log \frac{\hat{q}(s_{t+1}^{(1)}|s_{t+1}^{(0)})}{\hat{q}(s_{t+1}^{(n)}|s_{t+1}^{(0)})} + \log \prod_{k=2}^{n} \frac{p(s_{t+1}^{(k-1)}|s_{t+1}^{(k)}, s_t, a_t^k)}{\hat{q}(s_{t+1}^{(k-1)}|s_{t+1}^{(k)}, s_{t+1}^{(0)})} \right]
$$

$$
= \mathbb{E}_{\hat{q}(s_{t+1}^{(1):(n)}|s_{t+1})} \left[ \log \frac{p(s_{t+1}^{(n)})p(s_{t+1}^{(0)}|s_{t+1}^{(1)}, s_t, a_t^1)}{\hat{q}(s_{t+1}^{(n)}|s_{t+1}^{(0)})} + \sum_{k=2}^{n} \log \frac{p(s_{t+1}^{(k-1)}|s_{t+1}^{(k)}, s_t, a_t^k)}{\hat{q}(s_{t+1}^{(k-1)}|s_{t+1}^{(k)}, s_{t+1}^{(0)})} \right]
$$

Therefore, the evidence of dynamics transition can be bounded as follows:

$$
\log P(s_{t+1}|s_t, a_t^{1:N}) \geq \mathbb{E}_{\hat{q}(s_{t+1}^{(1)}|s_{t+1}^{(0)})}[\log p(s_{t+1}^{(0)}|s_{t+1}^{(1)}, s_t, a_t^1)] - \mathrm{D}_{\mathrm{KL}}[\hat{q}(s_{t+1}^{(n)}|s_{t+1}^{(0)})\|p(s_{t+1}^n)]
$$

$$
- \sum_{k=2}^{n} \mathbb{E}_{\hat{q}(s_{t+1}^{(k)}|s_{t+1}^{(0)})} \left[ \mathrm{D}_{\mathrm{KL}}(\hat{q}(s_{t+1}^{(k-1)}|s_{t+1}^{(k)}, s_{t+1}^{(0)})\|p(s_{t+1}^{(k-1)}|s_{t+1}^{(k)}, a_t^k, s_t)) \right]. \tag{13}
$$

As shown by [25], the conditional forward diffusion process $\hat{q}$ behaves identically to the unconditional one $q$. Therefore, we can substitute the $\hat{q}$ with the $q$ in Eq. (13), concluding our proof. $\square$

# B    EDM Preconditioners and Noise Scheduler

To keep input and output signal magnitudes fixed to the same scale and avoid large variance in gradient magnitudes on a per-sample basis, Karras et al. [21] introduced the following preconditioners for

normalization and re-scaling output to stabilize and improve the training dynamics of the network:

$$c_{\text{in}}^{\tau} = \frac{1}{\sqrt{\sigma(\tau)^2 + \sigma_{\text{data}}^2}} \tag{14}$$

$$c_{\text{out}}^{\tau} = \frac{\sigma(\tau)\sigma_{\text{data}}}{\sqrt{\sigma(\tau)^2 + \sigma_{\text{data}}^2}} \tag{15}$$

$$c_{\text{noise}}^{\tau} = \frac{1}{4}\log(\sigma(\tau)) \tag{16}$$

$$c_{\text{skip}}^{\tau} = \frac{\sigma_{\text{data}}^2}{\sigma_{\text{data}}^2 + \sigma(\tau)^2}, \tag{17}$$

where $\sigma_{\text{data}} = 0.5$ in our experiment hyperparameter setup. The noise scheduler for training the diffusion model follows the same design in [21], described as follows:

$$\sigma(\tau) = \tau, \ \log(\sigma) \sim \mathcal{N}(P_{\text{mean}}, P_{\text{std}}^2), \tag{18}$$

where $P_{\text{mean}} = -0.4$ and $P_{\text{std}} = 1.2$.

## C  Behavior Learning Details

Inspired by the success of MARIE [14], we adopt MAPPO [34] to train both the actor and critic inside the imaginations of DIMA. A key distinction from MARIE is that our model explicitly predicts the global state, enabling seamless integration with CTDE techniques as well as actor–critic architectures commonly used in model-free MARL. Therefore, we implement both the actor $\psi$ and critic $\xi$ with two 3-layer MLPs together with ReLU activation and Layer Normalization, respectively. Similar to off-the-shelf CTDE model-free MARL algorithms, we adopt actor parameter sharing across agents.

**Critic loss function.**  We utilize $\lambda$-return in DreamerV1 [1], which employs an exponentially weighted average of different $k$-steps TD targets to balance bias and variance as the regression target for the critic. Given an imagined trajectory $\{\hat{s}_t, \hat{o}_t^{1:n}, a_t^{1:n}, \hat{r}_t, \hat{\gamma}_t\}_{t=1}^H$ over all agents, $\lambda$-return is calculated recursively as,

$$V_\lambda(\hat{s}_t) = \hat{r}_t^i + \hat{\gamma}_t \cdot \begin{cases} (1-\lambda)V_\xi(\hat{s}_t) + \lambda V_\lambda(\hat{s}_{t+1}) & \text{if} \quad t < H \\ V_\xi(\hat{s}_t) & \text{if} \quad t = H \end{cases} \tag{19}$$

The objective of the critic $\xi$ is to minimize the mean squared difference $\mathcal{L}_\xi$ with $\lambda$-returns over imagined trajectories, as

$$\mathcal{L}_\xi = \mathbb{E}_{\pi_\psi}\left[\sum_{t=1}^{H-1}\left(V_\xi(\hat{s}_t) - \text{sg}\big(V_\lambda(\hat{s}_t)\big)\right)^2\right], \tag{20}$$

where $\text{sg}(\cdot)$ denotes the stop-gradient operation. We optimize the critic loss with respect to the critic parameters $\xi$ using the Adam optimizer.

**Actor loss function.**  The objective for the actor $\pi_\psi^i(\cdot|\hat{o}_t^i) := \pi_\psi(\cdot|\hat{o}_t^i)$ is to output actions that maximize the prediction of long-term future rewards made by the critic. To incorporate intermediate rewards more directly, we train the actor to maximize the same $\lambda$-return that was computed for training the critic. In terms of the non-stationarity issue in multi-agent scenarios, we adopt PPO updates, which introduce importance sampling for actor learning. The actor loss function for agent $i$ is:

$$\mathcal{L}_\psi^i = -\mathbb{E}_{\pi_{\psi_{\text{old}}}^i}\left[\sum_{t=0}^{H-1}\min\left(r_t^i(\psi)A_t, \text{clip}(r_t^i(\psi), 1-\epsilon, 1+\epsilon)A_t\right) + \eta\mathcal{H}(\pi_\psi^i(\cdot|\hat{o}_t^i))\right] \tag{21}$$

where $r_t^i(\psi) = \pi_\psi^i/\pi_{\psi_{\text{old}}}^i$ is the policy ratio and $A_t = \text{sg}(V_\lambda(\hat{s}_t) - V_\xi(\hat{s}_t))$ is the advantage. Unlike MAPPO, we choose not to design agent-specific global states, as such designs are overly hand-crafted and inject task-specific human priors, which undermines the generality and soundness of the approach. Instead, we retain the environment's original agent-agnostic global state shared among all agents, and feed it into the value function $V_\xi$. As a result, the estimated advantage function $A_t$ is also shared across all agents during actor updates. We optimize the actor loss with respect to the actor parameters $\psi$ using the Adam optimizer. Overall hyperparameters are shown in Table 4.

Table 4: Behaviour learning hyperparameters.

| Hyperparameter | Value |
|---|---|
| ***Common*** | |
| Imagination horizon ($H$) | 15 |
| $\lambda$ | 0.95 |
| Clipping parameter $\epsilon$ | 0.1 |
| | |
| ***MAMuJoCo*** | |
| Discount factor $\gamma$ | 0.99 |
| $\eta$ | 0.001 |
| | |
| ***Bi-DexHands*** | |
| Discount factor $\gamma$ | 0.95 |
| $\eta$ | 0.01 |

# D  Illustrations of Experimental Environments

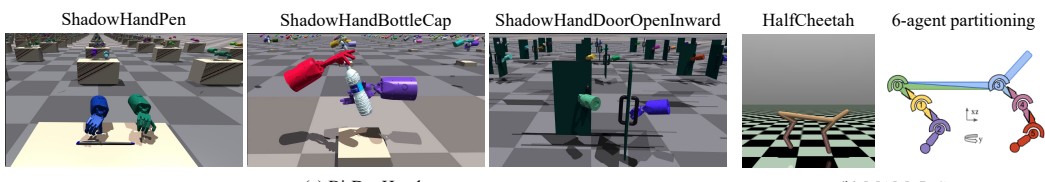

(a) Bi-DexHands
(b) MAMuJoCo

Figure 8: Illustrations of the experimental environments in our work. **Left**: Visualizations of three Bi-DexHands tasks: removing a pen cap, opening a bottle cap, and opening a door inwards. **Right**: Visualization of 6-agent partitioning w.r.t. HalfCheetah in MAMuJoCo.

# E   Additional Results

## E.1   Additional Experiments: Detailed Returns of All Methods on MAMuJoCo and Bi-DexHands

Table 5: **Comparison of final episode returns across MAMuJoCo and Bi-DexHands benchmarks.** We report the mean final episode return and standard deviation over 4 random seeds. DIMA consistently outperforms all baselines across all chosen tasks on both MAMuJoCo and Bi-DexHands. The best result per task is highlighted in bold and shaded in blue color, while the second-best is underlined.

| Tasks | Steps | Methods | | | | | |
|---|---|---|---|---|---|---|---|
| | | DIMA (Ours) | MARIE | MAMBA | HASAC | HAPPO | MAPPO |
| ***MAMuJoCo*** | | | | | | | |
| Ant-2x4 | | $\mathbf{4881}_{\pm756}$ | $\underline{4471}_{\pm553}$ | $1314_{\pm756}$ | $1344_{\pm282}$ | $1716_{\pm449}$ | $859_{\pm47}$ |
| Ant-4x2 | | $\mathbf{4766}_{\pm450}$ | $1173_{\pm136}$ | $1618_{\pm931}$ | $850_{\pm126}$ | $\underline{1917}_{\pm253}$ | $854_{\pm41}$ |
| HalfCheetah-2x3 | 1M | $\mathbf{6370}_{\pm121}$ | $\underline{4045}_{\pm275}$ | $2813_{\pm1580}$ | $2499_{\pm1081}$ | $2628_{\pm893}$ | $3196_{\pm75}$ |
| HalfCheetah-3x2 | | $\mathbf{6175}_{\pm212}$ | $2380_{\pm1145}$ | $3029_{\pm798}$ | $2872_{\pm890}$ | $\underline{3402}_{\pm317}$ | $2936_{\pm766}$ |
| HalfCheetah-6x1 | | $\mathbf{5643}_{\pm163}$ | $1738_{\pm1213}$ | $1848_{\pm220}$ | $2044_{\pm110}$ | $\underline{2939}_{\pm1113}$ | $925_{\pm121}$ |
| Walker2d-2x3 | | $\mathbf{3329}_{\pm1056}$ | $\underline{2822}_{\pm997}$ | $124_{\pm19}$ | $1135_{\pm210}$ | $1007_{\pm282}$ | $752_{\pm216}$ |
| Walker2d-3x2 | | $\mathbf{4084}_{\pm357}$ | $604_{\pm349}$ | $466_{\pm103}$ | $958_{\pm715}$ | $932_{\pm513}$ | $\underline{1004}_{\pm480}$ |
| ***Bi-DexHands*** | | | | | | | |
| BottleCap | | $\mathbf{259.9}_{\pm4.1}$ | - | $203.8_{\pm5.2}$ | $\underline{210.9}_{\pm6.1}$ | $100.7_{\pm3.8}$ | $104.0_{\pm2.3}$ |
| DoorOpenInward | 300K | $\mathbf{290.4}_{\pm29.0}$ | - | $225.0_{\pm79.4}$ | $\underline{246.3}_{\pm7.0}$ | $30.7_{\pm2.5}$ | $65.8_{\pm6.9}$ |
| DoorOpenOutward | | $\mathbf{367.1}_{\pm19.4}$ | - | $177.4_{\pm43.1}$ | $\underline{221.9}_{\pm7.3}$ | $58.8_{\pm4.6}$ | $96.4_{\pm8.5}$ |
| BottleCap | | $\mathbf{24.4}_{\pm11.4}$ | - | $\underline{4.3}_{\pm0.4}$ | $0.0_{\pm0.0}$ | $0.0_{\pm0.0}$ | $0.0_{\pm0.0}$ |

Table 6: Policy learning performance (final return) of DIMA with different imagination horizons ($H = 15$ vs. $H = 25$).

| Scenarios | DIMA w/ $H = 15$ | DIMA w/ $H = 25$ |
|---|---|---|
| Ant [4x2] (4-agent) | $\mathbf{4766}_{\pm450}$ | $4328_{\pm1058}$ |
| HalfCheetah [6x1] (6-agent) | $5643_{\pm163}$ | $\mathbf{6310}_{\pm335}$ |

## E.2   Additional Experiments: Policy Learning with Longer Imagination Horizons

We investigated the downstream impact on policy learning when utilizing the extended imagination horizon ($H = 25$) during the whole training process. The final policy returns are presented in Table 6. The results are environment-dependent and provide critical insights. On HalfCheetah [6x1], training with a longer horizon ($H = 25$) yields a substantial performance improvement. This strongly suggests that when DIMA's long-range predictions are stable, the policy optimizer can successfully leverage these extended rollouts to discover more complex and far-sighted strategies, leading to superior returns. Conversely, on Ant [4x2], we observe a performance degradation accompanied by significantly higher variance. This indicates that for this specific task, while comparatively lower than baselines (as shown in Table 1), the accumulated prediction errors in $H = 25$ rollouts are still sufficient to introduce noise that misleads the policy learning. This phenomenon underscores both the practical potential of DIMA in scalable model-based MARL (demonstrated on HalfCheetah) and highlights a key challenge: ensuring that the predictive accuracy is robust enough to provide a stable gradient signal for policy learning across all task types.

## E.3   Additional Experiments: Comparison on Training Compute

We measure the training time and GPU memory usage of all evaluated model-based MARL methods, including our proposed DIMA. All experiments were conducted using a single NVIDIA RTX 4090

GPU to ensure a fair comparison across methods and tasks. As shown in the table below, DIMA is substantially more efficient than MARIE in both training time and GPU memory usage, and is comparable to or slightly more efficient than MAMBA.

Table 7: Comparison on consumed computational resources over 2 test scenarios.

| Methods | Training Time | Usage of GPU Mem |
|---------|---------------|------------------|
| *Ant-2x4* | | |
| DIMA | 1d 19h | 3.10 GB |
| MARIE | 3d 17h | 14.33 GB |
| MAMBA | 1d 1h | 2.74 GB |
| *Ant-4x2* | | |
| DIMA | 1d 19h | 3.15 GB |
| MARIE | 3d 17h | 3.10 GB |
| MAMBA | 1d 1h | 4.37 GB |

### E.4 Ablation: Sequential vs. Joint Modeling Prediction Accuracy

Table 8: Ablation study comparing the **cumulative L1 observation errors** of sequential vs. joint modeling. Models were trained on 500k transitions and evaluated on a 500k held-out set. Sequential modeling achieves statistically indistinguishable prediction accuracy, validating its design.

| Task | Method | Obs L1 Error @ $H = 15$ | Obs L1 Error @ $H = 20$ |
|------|--------|--------------------------|--------------------------|
| DoorOpenOutward | Sequential (Ours) | $\mathbf{5.333}_{\pm 0.273}$ | $\mathbf{7.081}_{\pm 0.325}$ |
| | Joint | $5.345_{\pm 0.267}$ | $7.092_{\pm 0.324}$ |
| DoorOpenInward | Sequential (Ours) | $\mathbf{5.563}_{\pm 0.326}$ | $\mathbf{7.447}_{\pm 0.393}$ |
| | Joint | $5.565_{\pm 0.322}$ | $7.453_{\pm 0.386}$ |
| Pen | Sequential (Ours) | $\mathbf{6.667}_{\pm 1.764}$ | $\mathbf{8.936}_{\pm 2.328}$ |
| | Joint | $6.676_{\pm 1.762}$ | $8.947_{\pm 2.322}$ |

### E.5 Experiment Details: Imagination Evaluation across Different Conditioning Orders

To evaluate DIMA's imagination under different conditioning orders on the 2-agent Ant [2×4] task, we collect 10 episodes by using the final policy induced by our algorithm, and randomly sample 100 trajectory segments to form our trajectory segment dataset. For each segment, we generate imagined rollouts using DIMA with different action conditioning orders.

As the EDM framework decouples inference-time sampling from training, the number of denoising steps need not match the number of agents. Thus, we set the number of denoising steps equal to 4, i.e., twice the number of agents. Letting the agent set be $\{1, 2\}$, we consider three conditioning orders: (i) random order: $(2, 1, 1, 2)$, (ii) ascending order w.r.t. agent id: $(1, 1, 2, 2)$, and (iii) descending order w.r.t. agent id: $(2, 2, 1, 1)$.

To provide a quantitative evaluation in Figure 7 (right), we compute the L1 error per observation dimension at each timestep between the 100 sampled trajectory segments and their corresponding imagined rollouts, and accumulate the errors over the prediction horizon. All observation L1 errors are averaged across 2 agents.

## F  Overview of DIMA with Learning in Imaginations

We summarize the overall training procedure of DIMA paired with learning in imaginations in Algorithm 1 below. We denote as $\mathcal{D}$ the replay databuffer which stores data collected from the real environment.

# G  Training Details and Hyperparameters

## G.1  Model Architecture Details

**State decoder.**  To enable decentralized execution of policies trained within DIMA's imagination rollouts, each agent must make decisions based solely on its local observation rather than the shared global state. To support such policy learning, we introduce a necessary state decoder that maps the global state $s_t$ into the corresponding joint local observations $o_t^{1:n}$.

Due to our online model-based MARL setup, the state decoder must be continually updated under a non-stationary data distribution which also shifts continually. Using a vanilla MLP as the state decoder in this setting may lead to issues such as overfitting or mode collapse. To mitigate these risks, we incorporate additional regularization into the decoder design by adopting a Vector Quantized Variational Autoencoder (VQ-VAE) [30], which enforces a compact latent codebook representation via vector quantization. Among various VQ-VAE variants, we choose Finite Scalar Quantization (FSQ) [31] as our final implementation as it removes any auxiliary losses and achieves remarkably high codebook utilization, which indicates its strong and effective regularization.

Our implementation is based on the open-source repository: `https://github.com/lucidrains/vector-quantize-pytorch`. We simply build the encoder $E_\varphi$ and decoder $D_\varphi$ as MLPs to deal with continuous non-vision global states and joint local observations. The decoder is designed with the same hyperparameters as the encoder. The loss function for learning the autoencoder is as follows:

$$\mathcal{L}_{\text{FSQ}}(E_\varphi, D_\varphi) = \mathbb{E}_{(s_t, o_t^{1:n}) \sim \mathcal{D}} \left[ \| o_t^{1:n} - D_\varphi(E_\varphi(s_t) + \text{sg}(\text{round}(f(E_\varphi(s_t)))) - E_\varphi(s_t)) \|^2 \right],$$

(22)

where $f$ is a bounding function such that $i$-th channel/entry in $\hat{z}_t = \text{round}(f(E_\varphi(s_t)))$ takes one of $L_i$ unique values (here $f : z \to \lfloor L_i/2 \rfloor \tanh(z)$ for $i$-th channel/entry) and round is the operation to map real-valued inputs to the nearest integers. Therefore, we have an implicit codebook $\mathcal{C}$ with $|\mathcal{C}| = \prod_{i=1}^{d} L_i$. After training the VAE, our state decoder can be expressed by $g_\varphi(o_t^{1:n}|s_t) = D_\varphi(\text{round}(f(E_\varphi(s_t))))$. The hyperparameters are listed in Table 9.

**Diffusion model for dynamics modeling.**  We use the 1-D variant adapted from the U-Net 2D in DIAMOND [18] as the backbone of the diffusion model $D_\theta$. To predict the next state $s_{t+1}$, the diffusion model $D_\theta$ is initially conditioned on the current global $s_t$, joint action $a_t^{1:n}$ and the diffusion time $\tau$. To improve next-global-state prediction accuracy, we empirically augment the temporal context by additionally incorporating the last 2 global states and joint actions, extending it from $s_t$ and $a_t^{1:n}$ to $s_{t-2:t}$ and $a_{t-2:t}^{1:n}$. Note that the effect of sequential denoising is confined to the joint action $a_t^{1:n}$ conditioning at the current timestep $t$, and does not extend to the past joint actions.

Inspired by the success of DIAMOND, we directly adopt the same conditioning mechanism in DIAMOND, and use temporal stacking for global state conditioning and adaptive group normalization for joint action and diffusion time conditioning. The hyperparameters are listed in Table 9.

**Transformer as reward and termination model.**  The Transformer for predicting the reward and termination is built upon the implementation of minGPT [29]. Given a fixed imagination horizon $H$, it first takes a sequence of length $2H$ composed of global states and joint actions $(\ldots, s_t, a_t^{1:n}, \ldots)$, and encodes every single global state and joint action into $d_e$-dimensional embedding tensor via 2 separate encoding functions. Then the sequence tensor of shape $2H \times d_e$ is forwarded through fixed Transformer blocks. Finally, the Transformer predicts reward and termination via two separate 3-layer MLP heads $f_\phi(r_t|s_{\leq t}, a_{\leq t}^{1:n})$ and $f_\phi(\gamma_t|s_{\leq t}, a_{\leq t}^{1:n})$, respectively. In general, the loss function is described by

$$\mathcal{L}_\phi = \mathbb{E} \left[ \sum_{t=1}^{H} -\log f_\phi(r_t|s_{\leq t}, a_{\leq t}^{1:n}) - \log f_\phi(\gamma_t|s_{\leq t}, a_{\leq t}^{1:n}) \right].$$

(23)

But in practice, we optimize the reward prediction with a smooth L1 loss function and the termination prediction with a cross-entropy loss function. The hyperparameters are listed in Table 9.

## G.2  Computational Resources Used for Training

All our experiments including the evaluation of chosen baselines are run on a machine with a single NVIDIA RTX 4090 GPU, a 24-core CPU, and 256GB RAM.

### G.3 Baseline Implementation Details

In our experiments, we reran and evaluated all baseline methods. To ensure fairness for comparisons, we followed the optimal hyperparameters provided in their official implementations, listed below:

- MARIE: `https://github.com/breez3young/MARIE`;
- MAMBA: `https://github.com/jbr-ai-labs/mamba`;
- HASAC, HAPPO and MAPPO: `https://github.com/PKU-MARL/HARL`.

### G.4 DIMA hyperparameters

We list the hyperparameters of DIMA paired with learning in imaginations in Table 10.

## H Broader Impact

Our work introduces DIMA, a diffusion-inspired multi-agent world model that significantly improves sample efficiency in cooperative multi-agent control environments. By enabling more faithful imagined rollouts, DIMA can accelerate the development of complex autonomous systems—such as multiple real robots coordination, traffic management, and energy-efficient buildings—thereby reducing real-world trial costs. However, these capabilities also carry potential risks: misuse of high-fidelity neural simulators for adversarial planning could exacerbate privacy and security concerns.

**Algorithm 1:** DIMA paired with learning in imaginations

---

**Procedure** `training_loop()`:

    **for** *epochs* **do**

        `collect_experience(`*steps_collect*`)`

        **for** *steps_state_decoder* **do**

            `update_state_decoder()`

        **for** *steps_diffusion_model* **do**

            `update_diffusion_model()`

        **for** *steps_reward_end_model* **do**

            `update_reward_end_model()`

        **for** *steps_actor_critic* **do**

            `update_actor_critic()`

---

**Procedure** `collect_experience(`$n$`)`:

    $s_0, o_0^{1:n} \leftarrow$ `env.reset()`

    **for** $t = 0$ **to** $n - 1$ **do**

        Sample $a_t^i \sim \pi_\psi^i(a_t^i|o_t^i)$, $\forall$ agent $i$

        $s_{t+1}, o_{t+1}^{1:n}, r_t, \gamma_t \leftarrow$ `env.step(`$a_t^{1:n}$`)`

        $\mathcal{D} \leftarrow \mathcal{D} \cup \{s_{t+1}, o_{t+1}^{1:n}, a_t^{1:n}, r_t, \gamma_t\}$

        **if** $\gamma_t = 1$ **then**

            $s_{t+1}, o_{t+1}^{1:n} \leftarrow$ `env.reset()`

---

**Procedure** `update_state_decoder()`:

    Sample state-observation pair $(s_t, o_t^{1:n}) \sim \mathcal{D}$

    Compute $\mathcal{L}_{\text{FSQ}}$ in Eq. (22)

    Update State Decoder $g_\varphi$

**Procedure** `update_diffusion_model()`:

    Sample sequence $\big(s_{t-L+1}, a_{t-L+1}^{1:n}, \ldots, s_t, a_t^{1:n}, s_{t+1}\big) \sim \mathcal{D}$

    Sample $\log(\sigma) \sim \mathcal{N}(P_{\text{mean}}, P_{\text{std}}^2)$ and get $\tau = \sigma$ since $\sigma(\tau) := \tau$

    Sample $s_{t+1}^\tau \sim \mathcal{N}(x_{t+1}^0, \sigma^2 \mathbf{I})$

    Sample a chosen agent id $k \sim \text{Uniform}\{1, 2, \ldots, n\}$

    Compute $\hat{s}_{t+1}^{(0)} = D_\theta(s_{t+1}^\tau; \tau, a_t^k, \underbrace{s_{t-L+1:t}, a_{t-L+1:t-1}^{1:n}}_{\text{extra temporal context}})$

    Compute loss $\mathcal{L}(\theta) = \|\hat{s}_{t+1}^{(0)} - s_{t+1}\|^2$ in Eq. (9)

    Update Diffusion Model $D_\theta$

**Procedure** `update_reward_end_model()`:

    Sample sequence $\big(s_t, a_t^{1:n}, r_t, \gamma_t, \ldots, s_{t+H-1}, a_{t+H-1}^{1:n}, r_{t+H-1}, \gamma_{t+H-1}\big) \sim \mathcal{D}$

    **for** $i = t$ **to** $t + H - 1$ **do**

        Compute $\hat{r}_i \sim f_\phi(\hat{r}_i | s_{\leq i}, a_{\leq i}^{1:n})$ and $\hat{\gamma}_i \sim f_\phi(\hat{\gamma}_i | s_{\leq i}, a_{\leq i}^{1:n})$

    Compute $\mathcal{L}_\phi = \sum_{i=t}^{t+H-1} \text{CrossEntropy}(\hat{\gamma}_i, \gamma_i) + \text{SmoothL1}(\hat{r}_i, r_i)$ corresponding to Eq. (23)

    Update Reward and Termination Model $f_\phi$

**Procedure** `update_actor_critic()`:

    Set the joint action condition order $\rho = (i_1, i_2, \ldots, i_n)$

    Sample starting point $\big(s_{t-L+1}, o_{t-L+1}^{1:n}, a_{t-L+1}^{1:n}, \ldots, s_t, o_t^{1:n}\big) \sim \mathcal{D}$ of the imagination

    Let $\hat{o}_t^{1:n} = o_t^{1:n}$

    **for** $i = t$ **to** $t + H - 1$ **do**

        Sample $a_i^j \sim \pi_\psi^j(a_i^j | \hat{o}_i^j)$ $\forall$ agent $j$

        Sample the reward $\hat{r}_i$ and the termination $\hat{\gamma}_i$ with $f_\phi$

        Sample the next global state $\hat{s}_{t+1}$ by iteratively denoising with $D_\theta$ and $\rho$

        Sample the next joint observation state $\hat{o}_{t+1}^{1:n}$ with $g_\varphi$

    Update actor $\pi_\psi^i$ and critic $V_\xi$ via $\mathcal{L}_\xi$ and $\mathcal{L}_\psi^i$ over imaginations $\{\hat{s}_i, \hat{o}_i^{1:n}, a_i^{1:n}, \hat{r}_i, \hat{\gamma}_i\}_{i=t}^{t+H-1}$

Table 9: Architecture details.

| Hyperparameter | Value |
|---|---|
| **State Decoder** ($g_\varphi$) | |
| MLP layers | 3 |
| Hidden size | 512 |
| Activation | GELU [70] |
| FSQ Levels $L_i$ | [8, 6, 5] |
| | |
| **Diffusion Model** ($D_\theta$) | |
| Global state conditioning mechanism | Temporal stacking |
| Joint action conditioning mechanism | Adaptive Group Normalization |
| Diffusion time conditioning mechanism | Adaptive Group Normalization |
| Residual blocks layers | [2, 2, 2] |
| Residual blocks channels | [64, 64, 64] |
| Residual blocks conditioning dimension | 256 |
| | |
| **Reward and Termination Model** ($f_\phi$) | |
| Embedding dimension $d_e$ | 256 |
| Transformer block layers | 6 |
| Attention heads | 4 |
| Embedding dropout | 0.1 |
| Attention dropout | 0.1 |
| Residual dropout | 0.1 |

Table 10: Hyperparameters for DIMA.

| Hyperparameter | Value |
|---|---|
| Batch size for State Decoder training | 256 |
| Batch size for Diffusion Model training | 64 |
| Batch size for Reward and Termination Model training | 128 |
| Optimizer for State Decoder | AdamW |
| Optimizer for Diffusion Model | AdamW |
| Optimizer for Reward and Termination Model | AdamW |
| Optimizer for Actor & critic | Adam |
| Learning rate for State Decoder | 0.0003 |
| Learning rate for Diffusion Model | 0.0001 |
| Learning rate for Reward and Termination Model | 0.0001 |
| Learning rate for Actor & critic | 0.0005 |
| Gradient clipping for State Decoder | 10 |
| Gradient clipping for Diffusion Model | 1 |
| Gradient clipping for Reward and Termination Model | 10 |
| Gradient clipping for Actor & critic | 10 |
| Weight decay for State Decoder | 0.01 |
| Weight decay for Diffusion Model | 0.01 |
| Weight decay for Reward and Termination Model | 0.01 |
| $\lambda$ for $\lambda$-return computation | 0.95 |
| Discount factor $\gamma$ | see Table 4 |
| Entropy coefficient | see Table 4 |
| Buffer size (transitions) | $2.5 \times 10^5$ |
| Training steps per epoch | 200 |
| Training steps per epoch for policy learning | 4 |
| Sampling Environment steps per epoch | 200 in MAMuJoCo
500 in Bi-DexHands |
| PPO epochs | 5 |
| PPO Clipping parameter $\epsilon$ | 0.1 |
| Number of imagined rollouts | 600 |
| Imagination horizon $H$ | 15 |
| Diffusion sampling solver | Euler |
| Number of denoising steps | $\begin{cases} 2 \cdot \|\mathcal{N}\| \text{ if } \|\mathcal{N}\| \leq 2 \\ \|\mathcal{N}\| \text{ if } \|\mathcal{N}\| > 2 \end{cases}$ |

