# OpenReview forum: "Revisiting Multi-Agent World Modeling from a Diffusion-Inspired Perspective"
_NeurIPS.cc/2025/Conference — NeurIPS 2025 poster_

### Official Review · Reviewer_n42q · 2025-06-29

**Clarity:** 3
**Significance:** 2
**Originality:** 2
**Rating:** 4
**Confidence:** 4

**Summary:**

This manuscript proposes to model the multi-agent transition systems using a sequential diffusion model, where each action of each agent is denoised step by step instead of concatenated together. The authors develop a world model for MARL using diffusion models and empirically compared with baselines over some multi-agent benchmarks.

**Questions:**

Where is the proof of Thm. 2? It should be in the appendix but the readers can't be find it in the paper.

**Ethical Concerns:**

["NO or VERY MINOR ethics concerns only"]

**Final Justification:**

The reviewer has addressed my concerns on sample efficiency and comparison with previous joint modeling-based approaches.

**Limitations:**

Please see weakness.

**Paper Formatting Concerns:**

No concern.

**Quality:**

2

**Strengths And Weaknesses:**

Strengths:

1. The paper is easy to follow and well-written.
2. The paper conducts extensive experiments and compared with some baselines.

Weaknesses:

1. One major concern I have is that it's not clear to me why the proposed approach has superior sample efficiency. I don't think the final returns upon convergence have big difference with other baselines (e.g., HAPPO, if you check the results therein). The authors doesn't explain where is the sample efficiency from and no ablation study can help understand it either. I'm also curious about the comparison on the training cost (e.g., time, resources) for each method.

2. The second concern I have is the motivation of denoising each step instead of using "conventional approaches". In the section 5.4, the joint case and sequential case has almost the same expected returns, sample efficiency, and variance. The difference is very marginal. However, there are downsides of using sequential training, like the authors mention, permutation invariance and condition-independent noise process. In this case, being simpler and using the joint case is preferred.

---

> ### Author Rebuttal · Authors · 2025-07-30
>
> Dear Reviewer n42q,
>
> We sincerely appreciate your precious time and constructive comments. In the following, we would like to answer your concerns separately.
>
> **Weakness 1**: One major concern I have is that it's not clear to me why the proposed approach has superior sample efficiency. I don't think the final returns upon convergence have big difference with other baselines (e.g., HAPPO, if you check the results therein). The authors doesn't explain where is the sample efficiency from and no ablation study can help understand it either. I'm also curious about the comparison on the training cost (e.g., time, resources) for each method.
>
> **Response**: Thanks for the comment.
> 1. We would like to clarify a potential misunderstanding about the "sample efficiency" and "final return" in the context of our work.
> "Sample efficiency" refers to the ability of a policy to achieve higher performance with fewer samples (transitions) during training, indicating more effective sample utilization. Thus, in the low-data regime where policy training relies on a very limited number of samples, higher performance directly reflects better sample efficiency, as it demonstrates the model's ability to learn effective policies with limited data.
> In such context, "final return" refers to the performance at the end of training under this limited data budget, not the asymptotic return after sufficiently enough steps. Therefore, a higher final return under strict interaction constraints is itself an indicator of better sample efficiency.
> Illustrated in Figure 4, Figure 5 and Table 2 in Section E of the appendix, DIMA consistently outperforms all selected baselines in the low data regime, demonstrating superior sample efficiency.
> For convenience, we provide numerical results below (with bold for the highest value and underlined for the second highest). HASAC is a strong off-policy model-free MARL algorithm which can also leverage the experience collected by different policy.
> **When comparing DIMA’s performance at 1M steps to HASAC at 5M steps, DIMA achieves comparable or better returns with 5x fewer samples, demonstrating strong sample efficiency**. Moreover, DIMA shows more stable performance, as indicated by lower variance across runs.
>
> | Scenarios | HASAC @ 1M steps | HASAC @ 5M steps |  DIMA @ 1M steps |
> | :-: |:-:|:-:|:-:|
> | Ant 2x4 |1344 ± 282| $\underline{4830}$ ± 757| **4881** ± 756|
> | Ant 4x2  |850 ± 126| **5023** ± 753| $\underline{4766}$ ± 450|
> | HalfCheetah 6x1 |2044 ± 110| $\underline{5320}$ ± 2120| **5643** ± 163|
>
>
> 2. We measure the training time and GPU memory usage of all evaluated model-based MARL methods, including our proposed DIMA. All experiments were conducted using a single NVIDIA RTX 4090 GPU to ensure a fair comparison across methods and tasks. As shown in the table below, DIMA is substantially more efficient than MARIE in both training time and GPU memory usage, and is comparable to or slightly more efficient than MAMBA.
>
> | Methods | Training Time | Usage of GPU Mem |
> | :-: |:-:|:-:|
> | DIMA in Ant 2x4  | ~1d 19h |3.10 GB|
> | MARIE in Ant 2x4 | ~3d 17h |14.33 GB|
> | MAMBA in Ant 2x4 | ~1d 1h  |2.74 GB|
> | DIMA in Ant 4x2  | ~1d 19h |3.15 GB|
> | MARIE in Ant 4x2 | ~3d 17h |15.73 GB|
> | MAMBA in Ant 4x2 | ~1d 1h  |4.37 GB|
>
> **Weakness 2**: The second concern I have is the motivation of denoising each step instead of using "conventional approaches". In the section 5.4, the joint case and sequential case has almost the same expected returns, sample efficiency, and variance. The difference is very marginal. However, there are downsides of using sequential training, like the authors mention, permutation invariance and condition-independent noise process. In this case, being simpler and using the joint case is preferred.
>
> **Response**: We appreciate the reviewer’s comment. The core question addressed by our ablation is what advantages sequential modeling offers over joint modeling in multi-agent dynamics learning. The key benefit of sequential modeling lies in decomposing the exponentially large joint state and action space into a compositionally structured, linearly scaling space. This decomposition theoretically reduces the complexity of modeling multi-agent interactions, thereby enabling more sample-efficient learning of the world model, especially under the condition where limited data is available.
> We performed additional experiments on the ShadowHandBottleCap task in the Bi-DexHands environment, to examine whether our proposed sequential modeling brings reduced modeling complexity and further facilitates more efficient policy learning.
> The result on ShadowHandBottleCap is computed by averaging 8 independent runs, which is twice the number of runs used in the original ablation study reported in the main paper.
>
> | Methods | BottleCap @ 100K steps | BottleCap @ 150K steps | BottleCap @ 200K steps |
> | :-: |:-:|:-:|:-:|
> | joint |234.1 ± 20.6|238.6 ± 22.9|246.7 ± 10.9|
> | sequential |**251.8** ± 17.3|**248.2** ± 11.6|246.3 ± 14.6|
>
> Here, we also include the detailed final return results in Figure 8.
>
> | Methods | DoorOpenOutward @ 300K steps | DoorOpenInward @ 300K steps |
> | :-: |:-:|:-:|
> | joint |302.5 ± 76.9|235.1 ± 68.1|
> | sequential |**352.4** ($\uparrow$ 49.9) ± 40.5|**290.3** ($\uparrow$ 55.2) ± 30.4|
>
> The overall results consistently favor the sequential modeling across these evaluated tasks. Notably, in the more data-constrained setting (BottleCap @ 100K and 150K steps), sequential modeling yields higher sample efficiency.
>
> **As for the potential downsides of our proposed formulation**, we would like to clarify that permutation invariance is not strictly required. Using a fixed order during both training and inference can be enough in practice. However, this can introduce implicit bias, as the predicted next state may become coupled with a fixed order. In contrast, the ground-truth environment dynamics exhibits order-invariance by nature.
>
> To avoid this mismatch, we intentionally design DIMA to be permutation-invariant. In addition, the condition independence of the forward process is directly borrowed from the definition and derivation of conditional diffusion models in [1]. These design choices lead to a new objective (Eq. 9) as simple as the original one (Eq. 1), which we see as a strength rather than a limitation.
>
>
> **Question 1**: Where is the proof of Thm. 2? It should be in the appendix but the readers can't be find it in the paper.
>
> **Response**: The proof of Theorem 2 is provided in the **appendix PDF included in the supplementary material**, which was submitted alongside the main paper. We kindly ask the reviewer to refer to this file for the full derivation.
>
> **Reference**
>
> [1] Dhariwal, Prafulla, and Alexander Nichol. "Diffusion models beat gans on image synthesis." NIPS (2021): 8780-8794.

---

> > ### Comment · Reviewer_n42q · 2025-08-04
> > **Response to Authors' Rebuttal**
> >
> > I'd like to thank the reviewer for the detailed explanation, which is helpful for me to better understand the framework and also address my concerns.

---

> > > ### Author Response · Authors · 2025-08-04
> > > **Thank you for your feedback**
> > >
> > > Dear Reviewer n42q,
> > >
> > > We sincerely thank you for your positive feedback. We are glad that our response helped clarify the framework and address your concerns. Please feel free to reach out if any further questions arise.

---

### Official Review · Reviewer_Uzmx · 2025-06-29

**Clarity:** 3
**Significance:** 2
**Originality:** 3
**Rating:** 3
**Confidence:** 4

**Summary:**

The paper proposes DIMA as world model for multi-agent systems with diffusion techniques.
Different from traditional joint action condition in multi-agent world models, DIMA applies sequential action conditions with permutation invariance, to reduce the modeling complexity. This design is novel, while I expect to see stronger evidence that it is better than joint action conditions in terms of both computation, prediction accuracy and policy performance. The experiments on MAMuJoCo and Bi-Dexhands show better efficiency and final returns over prior model-based and model-free MARL baselines.

**Questions:**

Current world dynamics is modeled by diffusion, while reward and termination is modeled by transformer using MinGPT, is there any benefits for this separate modeling choice? and would it be possible to unify the modeling of different variables in one big diffusion model instead?

**Ethical Concerns:**

["NO or VERY MINOR ethics concerns only"]

**Final Justification:**

Thanks for the rebuttal and additional results. My concern on baseline is addressed. However, I still do not see statistically significant results that joint modeling is better than sequential modeling (which is the main technical contribution of the work), given the large fluctuation itself in MBRL process, therefore I keep my current score.

**Limitations:**

The paper should also discuss the computational cost for DIMA over other model-based or model-free MARL methods, since diffusion model rollout for multi-agent system can be quite heavy and RL policies usually require a large number of samples.

**Quality:**

2

**Strengths And Weaknesses:**

Strengths:

The paper is well written and easy to follow.

The design of sequential modeling for multi-agent action condition is novel, with some experiments showing comparable performances against joint action condition. This should save some modeling efforts due to the reduced complexity.

Experimental advantages over prior methods are clear.


Weakness:

For comparison of future trajectory prediction over different modeling methods (Fig. 6), please provide numerical results of observation and reward errors at each step over a long horizon.

The comparison of DIMA’s sequential modeling against the traditional joint agent action modeling, which is shown in Fig. 8, does not clearly tell the difference of the performances and the variances. The learning curves overall have high fluctuations, and it is not significant to see the lower variances by DIMA. Please prove this argument by more environments, and perhaps more runs to show with higher statistical confidence.

---

> ### Author Rebuttal · Authors · 2025-07-30
>
> Dear Reviewer Uzmx,
>
> We sincerely appreciate your precious time and constructive comments. In the following, we would like to answer your concerns separately.
>
> **Weakness 1**: For comparison of future trajectory prediction over different modeling methods (Fig. 6), please provide numerical results of observation and reward errors at each step over a long horizon.
>
> **Response**: Thanks for the constructive suggestion. Beyond the horizon $H = 15$ pre-defined during the training, We conduct additional evaluations with a longer prediction horizon of $H=25$, measuring accumulated observation and reward reconstruction errors over time. Here we still select the scenario *Ant-v2-2x4* for the experiment. The results are summarized below, and suggest that **DIMA exhibits lower compounding error trends over time compared to other baselines**.
>
> | Methods | Obs Accumulation Errors @ $H=15$ | Obs Accumulation Errors @ $H=25$ | Rew Accumulation Errors @ $H=15$ | Rew Accumulation Errors @ $H=25$ |
> | :-: |:-:|:-:|:-:|:-:|
> | DIMA |**2.42** ± 0.93|**4.32** ± 1.44|**15.01** ± 9.88|**31.07** ± 17.63|
> | MARIE |3.54 ± 1.60|6.62 ± 2.68|21.64 ± 17.51|47.66 ± 32.02|
> | MAMBA |3.98 ± 1.54|6.85 ± 2.51|38.90 ± 24.48|71.67 ± 41.78|
>
> Moreover, we would like to describe the detailed experiment setup of this error analysis. Since learning the world model is tied to a progressively improving policy both in MARIE, MAMBA and DIMA, we separately used their final policies to sample 5 episodes for fairness. We then computed L1 errors per observation between 100 trajectory segments randomly sampled from all 15 episodes and their imagined counterpart.
>
>
> **Weakness 2**: The comparison of DIMA’s sequential modeling against the traditional joint agent action modeling, which is shown in Fig. 8, does not clearly tell the difference of the performances and the variances. Please prove this argument by more environments, and perhaps more runs to show with higher statistical confidence.
>
> **Response**: Thanks for the constructive suggestion. To further clarify the performance differences between sequential modeling and joint modeling, we conducted additional experiments on the ShadowHandBottleCap task in the Bi-DexHands environment.
> Motivated by this comment, we took a closer look at the core question that our ablation study aims to answer—specifically what sequential modeling is fundamentally contributing to.
>
> The primary benefit lies in decomposing the exponentially expanded space of the multi-agent dynamics into a linearly scaling space, significantly reducing modeling complexity. This facilitates more efficient learning of the world model under more limited data, which in turn accelerates policy convergence and improves sample efficiency.
>
> To better highlight this, we calculate the averaged performance **under varying data budget** to test whether the reduced modeling complexity introduced by sequential modeling translates into improved policy performance.
> Notably, the result on ShadowHandBottleCap is computed by averaging 8 independent runs, which is twice the number of runs used in the original ablation study reported in the main paper.
>
> | Methods | BottleCap @ 100K steps | BottleCap @ 150K steps | BottleCap @ 200K steps |
> | :-: |:-:|:-:|:-:|
> | joint |234.1 ± 20.6|238.6 ± 22.9|246.7 ± 10.9|
> | sequential | **251.8** ± 17.3| **248.2** ± 11.6|246.3 ± 14.6|
>
> We observe that under low-data regimes (e.g., 100K steps and 150K steps), sequential modeling outperforms joint modeling in both average return and variance. As the data budget decreases, the performance gap widens. This trend demonstrates that in extremely low-data regimes, sequential modeling benefits from reduced modeling complexity, enabling more accurate world model learning and accelerating policy learning. Our method's ability to maintain strong performance even with severely limited data underscores its effectiveness in resource-constrained scenarios, which is the primary focus of our work.
>
> **Question 1**: Current world dynamics is modeled by diffusion, while reward and termination is modeled by transformer using MinGPT, is there any benefits for this separate modeling choice? and would it be possible to unify the modeling of different variables in one big diffusion model instead?
>
> **Response**: Thanks for the question.
> 1. We intentionally model the environment dynamics (i.e., state transitions) using a diffusion model, while predicting reward and termination separately via using a transformer-based architecture. This design choice stems from the inherent differences in the learning objectives of these variables. Note that reward prediction is typically learned using regression losses such as L2 loss, and termination prediction is often optimized via negative log-likelihood or binary cross-entropy loss. If all variables were modeled together in a single diffusion process, one would be forced to use a denoising loss, which is not well-aligned with the learning objectives for reward and termination. Therefore, we separate these components to better match their respective loss functions and learning dynamics.
>
> 2. Our choice of using a transformer (MinGPT) for modeling reward and termination is motivated by prior successes in Transformer-based world models. Notably, transformer-based world models such as MARIE [1], IRIS [2], and TWM [3] have demonstrated strong performance in capturing temporal dependencies in reward and termination signals. Inspired by these works, we adopt a similar design for these components.
>
> **Reference**
>
> [1] Zhang, Yang, et al. "Decentralized Transformers with Centralized Aggregation are Sample-Efficient Multi-Agent World Models." TMLR.
>
> [2] Micheli, Vincent, Eloi Alonso, and François Fleuret. "Transformers are Sample-Efficient World Models." ICLR 2023.
>
> [3] Robine, Jan, et al. "Transformer-based World Models Are Happy With 100k Interactions." ICLR 2023.

---

> > ### Comment · Reviewer_Uzmx · 2025-08-04
> >
> > I've read through the rebuttal. I wonder if the reward and termination transformer is motivated by IRIS and TWM, why these two are not taken as baseline methods (also for dynamics) if they are quite effective. TWM seems to also claim its effectiveness in low-data regime, therefore should be a quite important baseline method to compare with.
> >
> > For sequential versus joint modeling, it seems the additional results show that the low variances and high performances with sequential modeling is only effective in even lower-data regime (100k, and 150k), and actually worse in 200k case, and this ablation is not on originally claimed 300k regime. Since main results in paper is conducted with 300k data, the claim of "more stable final performance with reduced variance" in section 5.4 is actually invalid in such setting. This actually raises a concern, does sequential modeling by DIMA actually work worse than joint modeling with more data (larger or equal to 300k)?

---

> > > ### Author Response · Authors · 2025-08-05
> > > **Further response to reviewer Uzmx's concerns (Part 1)**
> > >
> > > Dear Reviewer Uzmx,
> > >
> > > Thank you for your thoughtful follow-up. We are happy to clarify the motivation behind our design choices and provide additional experimental evidence to address your concerns.
> > >
> > > 1. **On including IRIS and TWM as baselines**:
> > > While TWM and IRIS explore modeling reward and termination functions with Transformer architectures, our approach is more directly inspired by MARIE [1], which serves as a stronger baseline in the context of model-based MARL. Specifically, both IRIS [2] and TWM [3] are designed for **single-agent settings** with discrete action spaces, and are only evaluated under the Atari 100k benchmark. These settings diverge significantly from our multi-agent continuous control environments, making direct comparison less meaningful. In contrast, MARIE is a **Transformer-based multi-agent world model** that models dynamics via next-token prediction, incorporates the Centralized Training with Decentralized Execution (CTDE) principle into world modeling and achieves SOTA performance in learning in imaginations under a low data regime, making it a more appropriate baseline in our setting. Conceptually, MARIE can be seen as a multi-agent extension of IRIS.
> > >
> > > 2. **On the effectiveness of sequential vs. joint modeling**:
> > > We believe there may be a misunderstanding regarding the motivation and theoretical underpinning of our proposed diffusion-inspired sequential modeling. As shown in Theorem 2, we reformulate the one-step state transition (which should be an end-to-end prediction) into a **progressive denoising process**, which reduces the learning complexity of the world model from exponential to linear in the number of agents. This formulation is intended to reduce the modeling complexity during the world model learning, rather than guaranteeing better predictive performance than joint modeling when data is sufficient.
> > > Instead, we expect the proposed method with low-data regime to perform comparably well  with global model under sufficient-data regime.
> > >
> > > Crucially, when the ground-truth dynamics correspond to the Dec-POMDP transition $p(s_{t+1} | s_t, a_t^{1:n})$ (i.e., the target of joint modeling), our DIMA model optimizes an **evidence lower bound (ELBO)** on the log-likelihood of this joint transition. Therefore, when we have access to sufficiently enough data, joint and sequential modeling are theoretically expected to perform similarly in terms of predictive accuracy, and sequential modeling does not necessarily outperform joint modeling.
> > >
> > > To further support this and also address the reviewer's concern about the comparison result on BottleCap with more data (up to 300k steps), we report additional results for the **BottleCap** scenario under higher data budgets. As shown in the table below, at 200k–300k steps, sequential modeling achieves comparable performance and variance to joint modeling. Note that **BottleCap is a relatively easy task**, and as shown in Figure 5 of the main paper, sequential modeling already converges well at 100k steps. That is also why we chose lower data budgets for this scenario in the rebuttal. However, more challenging scenarios such as **DoorOpen** still benefit from increased training steps up to 300k. This task-specific treatment reflects differing low-data regimes depending on task difficulty.
> > >
> > > | Methods     | BottleCap @ 100K | BottleCap @ 150K | BottleCap @ 200K | BottleCap @ 250K | BottleCap @ 300K |
> > > |-------------|------------------|------------------|------------------|------------------|------------------|
> > > | Joint       | 234.1 ± 20.6     | 238.6 ± 22.9     | **246.7 ± 10.9** | 243.7 ± 18.2     | **255.2 ± 7.0**  |
> > > | Sequential  | **251.8 ± 17.3** | **248.2 ± 11.6** | 246.3 ± 14.6     | **251.9 ± 12.7** | 249.2 ± 10.7     |
> > >
> > > Together, these results align with our theoretical expectation: **Sequential modeling improves sample efficiency in the low-data regime**, while its performance converges to that of joint modeling when more data is available.

---

> > > > ### Author Response · Authors · 2025-08-05
> > > > **Further response to reviewer Uzmx's concerns (Part 2)**
> > > >
> > > > 3. Moreover, we would like to clarify the scope of our claim in Section 5.4 that "our proposed formulation leads to more stable final performance with reduced variance." This statement refers specifically to **limited data regimes**, where sequential modeling provides **reduced modeling complexity** and then improves **sample efficiency**, as evidenced by its **lower variance and higher return** at 100k and 150k steps. The results at 200k–300k steps are **not contradictory** to this claim; instead, they indicate that when sufficient data is available, both modeling strategies converge to similar levels of performance. Thus, the variance reduction benefit of sequential modeling is **more prominent in lower-data scenarios**, which is the context in which our claim holds.
> > > > In fact, our ablation study (Section 5.4) clearly states that *"DIMA achieves comparable overall performance to centralized world models."* Our focus is on showing that **the sequential formulation is an effective alternative when data is limited**, and we design our experiments to highlight this advantage.
> > > >
> > > > To further support this point, we also report the results of the main paper on two more challenging tasks, **DoorOpenOutward** and **DoorOpenInward**, where DIMA (i.e., sequential modeling) demonstrates **better policy performance and reduced variance** compared to the joint modeling baseline under a relatively strict data budget.
> > > > , even under the full 300k training steps. These results underscore that in **more complex tasks**, the structural bias introduced by our sequential formulation can remain beneficial:
> > > >
> > > > | Methods | DoorOpenOutward @ 300K steps | DoorOpenInward @ 300K steps |
> > > > | :-: |:-:|:-:|
> > > > | joint |302.5 ± 76.9|235.1 ± 68.1|
> > > > | sequential |**352.4** ($\uparrow$ 49.9) ± 40.5|**290.3** ($\uparrow$ 55.2) ± 30.4|
> > > >
> > > > Together, these results deliver our core claim: **Sequential modeling can provide better sample efficiency (i.e., better performance and reduced variance) in the limited data regime**.
> > > >
> > > > We hope this resolves your concerns and clarifies the intentions behind our formulation and claims.
> > > >
> > > > **Reference**
> > > >
> > > > [1] Zhang, Yang, et al. "Decentralized Transformers with Centralized Aggregation are Sample-Efficient Multi-Agent World Models." TMLR.
> > > >
> > > > [2] Micheli, Vincent, Eloi Alonso, and François Fleuret. "Transformers are Sample-Efficient World Models." ICLR 2023.
> > > >
> > > > [3] Robine, Jan, et al. "Transformer-based World Models Are Happy With 100k Interactions." ICLR 2023.

---

> > > > > ### Comment · Reviewer_Uzmx · 2025-08-08
> > > > >
> > > > > Thanks for the rebuttal.
> > > > >
> > > > > I understand that the main claim of the paper is that sequential modeling is better than joint modeling due to reduced modeling complexity at low data regime. And the original claimed low data regime is 300k data, where the performance difference of the two is not clear to me at 200 to 300k data. Even current 100-200k results does not tell a clear advantage of joint modeling given the large confidence margin. Given the large fluctuation of learning curves of MBRL setting, I cannot tell sequential modeling is stastically significant better than joint modeling. A better way to prove this would just be the comparison of prediction errors of observation and reward, over multiple (5-10) tasks. Also, if the reduced modeling complexity is indeed effective, it should be more obvious to see a clear dominating learning curves before 100-200k steps, but this is not shown in Fig. 8.

---

> > > > > > ### Author Response · Authors · 2025-08-08
> > > > > > **Thank you for insightful comment**
> > > > > >
> > > > > > Dear Reviewer Uzmx,
> > > > > >
> > > > > > We truly appreciate your insightful suggestion of *using prediction error as a direct metric* to compare the two world model formulations. We consider this a valuable idea, and it is very helpful in refining our evaluation approach. Our previous analysis focused primarily on comparing the performance of the policy learned inside the world model, which is instead an indirect measure and can be influenced by many confounding factors--such as stochastic exploration triggered by policy stochasticity, environment stochasticity, and updates from the RL algorithm. These factors partly contribute to the high fluctuation observed in MBRL learning curves and may obscure the underlying modeling differences.
> > > > > >
> > > > > > Following your advice, we plan to conduct additional experiments to compare the observation and reward prediction errors of the two modeling approaches on three Bi-DexHands scenarios--*DoorOpenInward*, *DoorOpenOutward* and *BottleCap*--under varying data budgets. We believe this direct evaluation will more clearly isolate and reveal the effect of reduced modeling complexity. If the experiments can be completed within the remaining discussion period, we will promptly report the new results.

---

### Official Review · Reviewer_DfzB · 2025-07-03

**Clarity:** 3
**Significance:** 3
**Originality:** 2
**Rating:** 4
**Confidence:** 3

**Summary:**

- The paper is about modeling environment dynamics in MARL using a diffusion process.
- The dynamic modeling problem is formulated as a diffusion-based conditional denoising process, which is claimed to reduce complexity without requiring additional communication mechanisms.
 - The key idea is to shift from modeling joint state-action transitions directly to modeling only the state transitions step by step, conditioning progressively on each agent’s action.
 - Experiments on MAMuJoCo and Bi-DexHands benchmarks demonstrate that DIMA outperforms existing model-based MARL baselines (such as MAMBA and MARIE) as well as some model-free methods (MAPPO, HAPPO, and HASAC).

**Questions:**

- Can you clarify the practical implication of Theorem 2 for the main algorithm?
- How realistic is Assumption 1? Could you provide an example showing how this assumption holds in the multi-agent environments under consideration?

Please also respond to the specific points raised in the Weaknesses section above.

**Ethical Concerns:**

["NO or VERY MINOR ethics concerns only"]

**Final Justification:**

The authors have addressed most of my concerns. I believe the paper makes a meaningful contribution, and I maintain my evaluation as a borderline acceptance.

**Limitations:**

- Yes, the authors briefly mention a limitation regarding scalability. However, there might be other limitations, such as the need for sufficiently large and diverse datasets to accurately model the environment dynamics.
-  I do not see any major potential negative societal impact.

**Paper Formatting Concerns:**

The paper format seems find.

**Quality:**

3

**Strengths And Weaknesses:**

**Strengths**

- The connection between sequential agent modeling and diffusion processes is conceptually interesting.
 - DIMA compresses the information space from
∣S∣×∣S∣×∣A∣ to ∣S∣, which significantly reduces model complexity.
-  Results on challenging tasks like Bi-DexHands and various MAMuJoCo partitions (Figures 4–5) show clear improvements in sample efficiency and stability over strong baselines.
- The paper includes long-horizon prediction reconstructions (Figure 6), permutation invariance tests (Figure 7), and an ablation comparing sequential vs. joint agent modeling (Figure 8), which make the empirical analysis quite complete.

**Weaknesses**

- While I appreciate the idea of reducing the information space from
∣S∣×∣S∣×∣A∣ to  ∣S∣, I find this relatively simple and not very innovative.
- The paper mentions MADiff and MADiTS as relevant diffusion-based works but does not sufficiently compare with them beyond qualitative differences.
- While MAMuJoCo and Bi-DexHands are good benchmarks, adding an additional environment (such as SMACv2 [1]) could demonstrate the generality of the approach for more discrete-action cooperative tasks.
- Theorem 2 seems to be a key theoretical result in the paper, but its practical implication for the overall algorithm is not clear. I am wondering if the bound is trivial and does not actually help guide the main algorithm.
- There are some more recent model-free MARL algorithms that are worth mentioning and possibly including in the comparison (e.g., IMAX-PPO [2] ) as well as some transformer-based MARL approaches (e.g. MARIE).

**Refs**
[1] Ellis, B., Cook, J., Moalla, S., Samvelyan, M., Sun, M., Mahajan, A., Foerster, J. N., & Whiteson, S. (2023). SMACv2: An improved benchmark for cooperative multi-agent reinforcement learning. In Advances in Neural Information Processing Systems

[2]  Bui, T. V., Mai, T., & Nguyen, T. H. (2024). Mimicking To Dominate: Imitation Learning Strategies for Success in Multi-Agent Games. In Advances in Neural Information Processing Systems 37 (NeurIPS 2024)

---

> ### Author Rebuttal · Authors · 2025-07-30
>
> Dear Reviewer DfzB,
>
> We sincerely appreciate your precious time and constructive comments. In the following, we would like to answer your concerns separately.
>
> **Weakness 1**: While I appreciate the idea of reducing the information space from ∣S∣×∣S∣×∣A∣ to ∣S∣, I find this relatively simple and not very innovative.
>
> **Response**: Thank you for the comment. While our approach may appear relatively simple, we believe that its strength lies in this simplicity. By reducing the modeling space from $|\mathcal{S} | \times |\mathcal{A}|^n \times |\mathcal{S}|$ to $|\mathcal{S} | \times |\mathcal{A}| \times |\mathcal{S}|$, we are able to significantly lower the complexity of multi-agent dynamics modeling without sacrificing performance (see our ablation study in Section 5.4).
> This is achieved through a principled formulation of multi-agent prediction as a sequentially conditional denoising process, which enables centralized modeling without requiring explicit agent communication. Despite its simplicity, DIMA yields strong empirical gains across challenging benchmarks and achieves higher sample efficiency than prior model-free and model-based methods. We view this as an instance of a simple yet effective design, which we believe is valuable for the community.
>
> **Weakness 2**: The paper mentions MADiff and MADiTS as relevant diffusion-based works but does not sufficiently compare with them beyond qualitative differences.
>
> **Response**: Thanks for the constructive comment. As stated in Section 4 of our manuscript, MADiff [1] and MADiTs [2] differ from our method not only in their design but also in the specific research questions they aim to address. Consequently, they cannot serve as suitable baselines for our work. Specifically,
> 1. MADiff leveraged a diffusion model as a goal-conditioned trajectory generator and generated actions via an inverse dynamics model, which can be viewed as a multi-agent extension of DecisionDiffuser [3].
> 2. MADiTs used diffusion models for offline trajectory stitching, generating synthetic transitions to augment the original offline dataset.
>
> Crucially, both MADiff and MADiTs are **offline model-based MARL algorithms** that rely on high-quality offline data (e.g., near-optimal trajectories) to perform well. In contrast, **DIMA operates in the online model-based MARL setting**, where the data buffer is initialized empty and the diffusion-based world model is learned entirely from scratch during training, without any access to pre-collected expert demonstrations.
> Furthermore, MADiff and MADiTs use diffusion models to model goal-conditioned trajectory distributions, i.e., $p(o_t^{1:n}, o_{t+1}^{1:n}, ..., o_{t+H}^{1:n} | g)$, where the diffusion model serves primarily as a planner conditioned on a high-level goal $g$. In contrast, DIMA directly models the multi-agent transition dynamics, i.e., $p(s_{t+1} | s_{t-k:t}, a_{t-k:t}^{1:n})$ and $p(o_t^{1:n} | s_t)$, which is essential for agents' interactions in imagined environment.
>
> **Due to these fundamental differences in training setting (offline vs. online), modeling objective (planning vs. dynamics modeling), and usage of the diffusion model, MADiff and MADiTs are not directly comparable to DIMA under our evaluation protocol focused on learning in imagination.**
>
> **Weakness 3**: While MAMuJoCo and Bi-DexHands are good benchmarks, adding an additional environment (such as SMACv2) could demonstrate the generality of the approach for more discrete-action cooperative tasks.
>
> **Response**: Thanks for the suggestion. The primary contribution of the proposed DIMA lies in introducing a novel multi-agent world model that achieves improved accuracy and reduced complexity. In light of this, we conducted extensive experiments comparing DIMA against baselines across two continuous control benchmarks with 10 scenarios. To the best of our knowledge, prior model-based MARL methods quite struggle in these continuous control benchmarks. Since the continuous action space is more complex than the discrete one, these continuous control tasks can be considered much more difficult.
> In these experiments, DIMA consistently outperformed the baselines. The selected benchmarks span a variety of tasks, with the number of agents ranging from 2 to 6. This broad selection of tasks sufficiently demonstrates DIMA’s superior performance and general applicability.
> SMACv2, on the other hand, primarily focuses on environments with random agent types. When we decompose the original dynamics transition into a sequentially-conditioned structured transition, we also implicitly assume a fixed agent type for each agent ID throughout training, making SMACv2 incompatible with DIMA’s formulation.
> Thus, we chose not to include SMACv2 in our evaluation.
>
> **Weakness 4**: Theorem 2 seems to be a key theoretical result in the paper, but its practical implication for the overall algorithm is not clear. I am wondering if the bound is trivial and does not actually help guide the main algorithm.
>
> **Response**: Thanks for raising this important point. Theorem 2 provides a theoretical justification for our training objective in the case of a sequentially conditioned diffusion-based world model under Assumption 1. Specifically, the denoisng term in Eq. 7 derives a denoising loss that guarantees the model learns the correct conditional transition distribution.
> Without this result, it would be unclear how to formulate a principled objective to enforce the desired temporally sequential conditioning structure. Therefore, Theorem 2 plays an essential role in grounding our training procedure and ensuring the model’s behavior aligns with the intended design.
>
> **Weakness 5**: There are some more recent model-free MARL algorithms that are worth mentioning and possibly including in the comparison (e.g., IMAX-PPO) as well as some transformer-based MARL approaches (e.g. MARIE).
>
> **Response**: Thanks for the constructive suggestion.
> Regarding MARIE [4], we have already included it as a baseline in our experiments. As for IMAX-PPO [5], we were unable to find any publicly available implementation. As an alternative, we included HATD3 [6], an off-policy model-free MARL algorithm that can leverage experiences collected from different policies.
> The results under a low data regime (limiting real-environment samples to 1M for MAMuJoCo) is listed as follows. DIMA still significantly outperforms HATD3.
> | Scenarios | HATD3 @ 1M steps | DIMA @ 1M steps |
> | :-: |:-:|:-:|
> | Ant 2x4  | 1751 ± 444 | **4881** ± 756 |
> | Ant 4x2  | 1639 ± 346 | **4766** ± 450 |
> | HalfCheetah 2x3 | 2405 ± 1119 | **6370** ± 121 |
> | HalfCheetah 3x2 | 2406 ± 1255 | **6175** ± 212 |
> | HalfCheetah 6x1 | 1823 ± 770  | **5643** ± 163 |
>
> **Question 1**: Can you clarify the practical implication of Theorem 2 for the main algorithm?
>
> **Response**: Thanks for the question. Please see the **response to the Weakness 4**.
>
> **Question 2**: How realistic is Assumption 1? Could you provide an example showing how this assumption holds in the multi-agent environments under consideration?
>
> **Response**: We appreciate this insightful question. We illustrate Assumption 1 with two examples from multi-agent firefighting scenarios.
>
> First, consider three firefighters independently extinguishing fires at different locations. Each denoising step receives one agent’s extinguishing fire action and removes the uncertainty corresponding to that specific fire (i.e., part of the global state). The remaining uncertainty depends on the actions of the other agents, which are revealed in later steps. This aligns with the case that each agent’s action partially denoises the global state in a dimension-wise manner.
>
> Second, consider all three firefighters working together at the same location.
> When an agent take extinguishing fire action, it would slightly reduces the fire, progressively denoising the shared state. The final outcome depends on the combined effect of all actions, as the denoising steps cumulatively reduce uncertainty of the whole global state.
>
> **Limitation 1**: However, there might be other limitations, such as the need for sufficiently large and diverse datasets to accurately model the environment dynamics.
>
> **Response**: Thanks for the comment. As mentioned earlier, **DIMA operates in an online model-based MARL setting**, where the data buffer is initialized empty and the diffusion-based world model is learned entirely from scratch—without access to any pre-collected expert demonstrations.
> And our experimental results show that DIMA can learn an accurate enough world model to effectively accelerate policy learning under a low-data regime.
> Compared with prior online methods like MARIE and MAMBA which also focus on learning in imaginations paradigm, DIMA achieves superior performance with fewer environment interactions.
> In contrast, offline model-based MARL algorithms like MADiff and MADiTs instead demand a sufficiently large dataset to deliver good performance.
>
> **Reference**
>
> [1] Zhu, Zhengbang, et al. "MADiff: Offline multi-agent learning with diffusion models." NIPS (2024): 4177-4206.
>
> [2] Yuan, Lei, et al. "Efficient multi-agent offline coordination via diffusion-based trajectory stitching." ICLR 2025.
>
> [3] Ajay, Anurag, et al. "Is Conditional Generative Modeling all you need for Decision Making?." ICLR 2023.
>
> [4] Zhang, Yang, et al. "Decentralized Transformers with Centralized Aggregation are Sample-Efficient Multi-Agent World Models." TMLR.
>
> [5] Mai, Tien, and Thanh Nguyen. "Mimicking to dominate: Imitation learning strategies for success in multiagent games." NIPS (2024): 84669-84697.
>
> [6] Zhong, Yifan, et al. "Heterogeneous-agent reinforcement learning." JMLR (2024): 1-67.

---

> > ### Comment · Reviewer_DfzB · 2025-08-01
> >
> > I thank the authors for their response, which effectively addresses my concerns.
> >
> > The response also highlights a limitation of the current work: DIMA’s formulation appears unable to handle the complexity of SMACv2—a widely used benchmarking environment for evaluating MARL algorithms.

---

> > > ### Author Response · Authors · 2025-08-02
> > > **Thank you for your feedback**
> > >
> > > Dear Reviewer DfzB,
> > >
> > > We are glad to hear that our response effectively addressed your concerns. We appreciate your comment and would like to offer further clarification from the perspective of model-based MARL. While SMACv2 is a well-known benchmark in the MARL community, it has not been widely adopted in model-based MARL yet, and existing attempts often show limited success:
> > >
> > > 1. MADiff [1] an offline model-based method, performs significantly worse than model-free baselines on SMACv2 (see its Appendix H.1). MADiTs [2] includes SMACv2 but does not compare against strong offline model-free baselines (also see its Appendix H.1).
> > >
> > > 2. MARIE [3], which shares a same learning paradigm as ours, includes limited SMACv2 evaluation, with only marginal improvements over QMIX [4] which is a strong model-free value-based method. MAMBA [5] omits SMACv2 entirely.
> > >
> > > 3. We think the core reason why limited evaluation was conducted on SMACv2 in the context of model-based MARL is that SMACv2 randomizes unit types and agent positions per episode, implicitly turning each scenario into a multi-task setting. To our knowledge, none of the existing model-based MARL methods, including MADiff, MADiTs, MARIE, MAMBA, or our proposed DIMA, explicitly target for and address this setting.
> > >
> > > 4. Furthermore, although SMACv2 contains 15 scenarios, most model-based works only evaluate on a small subset. For example, MADiff was evaluated only on terran_5_vs_5, MADiTs on terran_5_vs_5 and zerg_5_vs_5, and MARIE on just 3 scenarios.
> > >
> > > We believe this reflects a current limitation of model-based MARL methods, and we view **extending such models to handle multi-task or even meta-learning settings as an important future direction**.
> > >
> > > **Reference**
> > >
> > > [1] Zhu, Zhengbang, et al. "MADiff: Offline multi-agent learning with diffusion models." Advances in Neural Information Processing Systems 37 (2024): 4177-4206.
> > >
> > > [2] Yuan, Lei, et al. "Efficient multi-agent offline coordination via diffusion-based trajectory stitching." The Thirteenth International Conference on Learning Representations. 2025.
> > >
> > > [3] Zhang, Yang, et al. "Decentralized Transformers with Centralized Aggregation are Sample-Efficient Multi-Agent World Models." Transactions on Machine Learning Research.
> > >
> > > [4] Rashid, Tabish, et al. "QMIX: Monotonic Value Function Factorisation for Deep Multi-Agent Reinforcement Learning." ICML. PMLR, 2018.
> > >
> > > [5] Egorov, Vladimir, and Alexei Shpilman. "Scalable Multi-Agent Model-Based Reinforcement Learning." AAMAS (2023).

---

### Official Review · Reviewer_Y1VU · 2025-07-03

**Clarity:** 3
**Significance:** 3
**Originality:** 3
**Rating:** 4
**Confidence:** 2

**Summary:**

The paper proposes a DIMA, a Diffusion-Inspired Multi-Agent world model to support training of multi-agent reinforcement learning policies.

The authors consider a Dec-POMDP (decentralized partially observable Markov decision process) where N agents interact in the same environment: given a state s_t, each agent i takes an action a^i_t given its local observation o^i_t of the state. Then, the environment moves from the current state s_t to the next state s_t+1, given the joint actions a_t, from all the agents.
Given the high complexity  of the joint action space A --- which increases exponentially with the number of agents and actions --- the authors propose a diffusion inspired module (DIMA) that scale linearly wrt the state space dimension. The main and novel idea is to leverage the denoising capability of diffusion models: starting from a noisy state representation s^0_t the module iteratively  applies the actions (a^i) from the agents, until all the actions  have been executed and the state have been denoised (thus we obtain the final state s_t).

Experimental results on two benchmarks support the claims: DIMA outperforms state-of-art approaches, providing higher episodic returns with fewer steps.

**Questions:**

Q1: How does the model handle long prediciton horizions? What is the impact of an increasing state/action space? How does the model interpolate or generalize to previously unseen actions/states?

Q2: what are the performance of the tested models using more steps / at convergence (e.g., 1e7 steps)?

Q3: what is the computational cost of training the world model (e.g., time), compared to similar approaches?

**Ethical Concerns:**

["NO or VERY MINOR ethics concerns only"]

**Final Justification:**

The paper provides a novel perspective and approach to support training of multi-agent reinforcement learning policies.

**Limitations:**

Yes - additional details would be helpful

**Quality:**

3

**Strengths And Weaknesses:**

Strengths:

-	The paper is well written and easy to follow. I like Fig.1, it provides a clear understanding of the approach.

-	The idea of using the diffusion denoising approach is novel. The denoising is used to predict the next state while handling the joint actions.

-	The results are interesting, they show better performance on the two tested benchmarks.


Weakness:

-	The scalability of the approach is not sufficiently tested. The evaluated prediction/ imagination horizon is limited, with H=15. This does not provide insights on the scalability of DIME to longer horizon. This is particularly important, as a well-known issue in auto-regressive world models is the accumulation of compounding errors, which can lead to unrealistic predictions (out-of-distribution).

-	The main paper lacks details or intuition about the architecture of the diffusion model. How does it handle (discrete?) finite actions and states?

-	The experiments have a limited number of steps (10^5 or 10^6 steps). For example, the cited HASAC paper studied the algorithm using up to 1e7 steps. I understand the approach have faster convergence, but what are the performance at convergence? I believe this experiment would be interesting for the reader.

-	The computational cost and training time of the world-agent are not investigated (also in comparison wrt similar solution).

Minor and typos:

- does the authors use “§” to refer to the appendix? If so, this should be stated explicitly at least once, for clarity.

---

> ### Author Rebuttal · Authors · 2025-07-30
>
> Dear Reviewer Y1VU,
>
> We sincerely appreciate your precious time and constructive comments. In the following, we would like to answer your concerns separately.
>
> **Weakness 1**: The scalability of the approach is not sufficiently tested.
>
> **Response**: We fully agree that evaluating the scalability of the world model—especially under longer imagination horizons—is essential, due to the well-known issue of compounding errors in predicting future observation with an auto-regressive manner. These errors can lead to significant deviations from the true dynamics over time, and addressing them is key to ensuring reliable imagined rollouts for downstream policy learning.
> As we provide a novel perspective to reduce the modeling complexity on learning multi-agent dynamics, it can implicitly enable DIMA to more accurately capture the underlying dynamics while demanding less data.
>
> To further assess the scalability of DIMA, we conduct additional evaluations with a longer prediction horizon of $H=25$, measuring accumulated observation reconstruction errors over time. Here we still select the scenario *Ant-v2-2x4* for the experiment. The results are summarized below:
> | Methods | Obs Accumulation Errors @ $H=15$ | Obs Accumulation Errors @ $H=25$ |
> | :-: |:-:|:-:|
> | DIMA |**2.42** ± 0.93|**4.32** ± 1.44|
> | MARIE |3.54 ± 1.60|6.62 ± 2.68|
> | MAMBA |3.98 ± 1.54|6.85 ± 2.51|
>
> We also evaluate policy learning performance under the extended imagination horizon, comparing DIMA with $H=15$ and $H=25$. The results are reported as follows.
> | Scenarios | DIMA w/ $H=15$ | DIMA w/ $H=25$ |
> | :-: |:-:|:-:|
> | Ant 4x2 (4-agent) |4766 ± 450| 4328 ± 1058 |
> | HalfCheetah 6x1 (6-agent) |5643 ± 163| 6310 ± 335 |
>
> **In conclusion**, these results suggest that DIMA not only exhibits lower compounding error trends over time compared to other baselines, but also sustains effective policy learning under a longer imagination horizon, thereby demonstrating its scalability.
>
> **Weakness 2**: The main paper lacks details or intuition about the architecture of the diffusion model. How does it handle (discrete?) finite actions and states?
>
> **Response**: Thank you for the constructive feedback. We provide detailed descriptions of our diffusion model architecture in Section G of the appendix (available in the supplementary materials). Briefly, our model adopts a 1D variant of the 2D U-Net architecture used in DIAMOND [1]. The global state is first pre-normalized by a running mean and standard deviation computed throughout training, and corrupted by adding noise for inputting the diffusion models.
> Regarding the joint actions, in order to achieve a sequentially conditioned reverse diffusion process, we employ an agent-wise mask to mask out the unused agents, and use a Transformer encoder to extract the feature from the action information in temporal context $a_{t-k:t-1}^{1:n}$ and current action $a_t^i$ of selected agent $i$. These actions are first encoded into embeddings via an MLP, and then processed by the Transformer encoder. The resulting representation is used as a conditioning input to the diffusion model.
> Given that all evaluated environments feature continuous actions, we employ an MLP to project the raw action inputs into embeddings. For tasks with discrete action spaces, an embedding layer would be a natural alternative. Regarding the discrete state, no modification is required for our method.
>
> **Weakness 3**: The experiments have a limited number of steps (10^5 or 10^6 steps). For example, the cited HASAC paper studied the algorithm using up to 1e7 steps.
>
> **Response**: Thanks for the insightful comment.
> 1. The focus of our paper is on evaluating sample efficiency brought by a learned world model. **To this end, we intentionally adopt a low-data regime to assess how effectively different methods can learn under constrained data budgets. This setting is practically relevant for scenarios where data collection is expensive or limited, and where fast convergence is crucial.** Note that the "sample efficiency" within the context of RL refers to the ability of a policy to achieve higher performance with fewer samples (transitions) during training, indicating more effective sample utilization.
> 2. To address the reviewer’s concern regarding final performance at convergence, we additionally run HASAC for up to $10^7$ steps on selected scenarios. Specifically, we focus on the MAMuJoCo suite (e.g., Ant 2×4, Ant 4×2, and HalfCheetah 6×1), where performance continues to improve beyond 1M steps. In contrast, for the Bi-DexHands environments, HASAC exhibits early convergence to suboptimal policies around 300K steps (see Figure 5), making further training less beneficial.
>
> The extended results are summarized in the table below (bold for the highest value, underlined for the second highest).
> | Scenarios | HASAC @ 1M steps | HASAC @ 5M steps | HASAC @ 10M steps | DIMA @ 1M steps |
> | :-: |:-:|:-:|:-:|:-:|
> | Ant 2x4 |1344 ± 282|4830 ± 757|**5253** ± 617| $\underline{4881}$ ± 756|
> | Ant 4x2  |850 ± 126| $\underline{5023}$ ± 753|**5338** ± 667|4766 ± 450|
> | HalfCheetah 6x1 |2044 ± 110|5320 ± 2120|**6419** ± 2564| $\underline{5643}$ ± 163|
>
> According to the theoretical analysis in [2], policies optimized under an approximate dynamics model would exhibit a certain suboptimality compared to those trained with true dynamics. Although our policy is learned in a learned dynamics model rather than the ground-truth environment, its performance remains competitive. Meanwhile, as shown in Figures 4 and 5, the policy learning curve in DIMA continues to improve and has not yet converged, indicating potential for further gains.
>
> Furthermore, when comparing DIMA's performance at 1M steps to HASAC's performance at 5M steps, we observe that DIMA achieves higher returns with 5x fewer samples, highlighting its strong sample efficiency.
>
> **Weakness 4**: The computational cost and training time of the world-agent are not investigated (also in comparison wrt similar solution).
>
> **Response**: We measure the training time and GPU memory usage of all evaluated model-based MARL methods, including our proposed DIMA. All experiments were conducted using a single NVIDIA RTX 4090 GPU to ensure a fair comparison across methods and tasks.
>
> | Methods | Training Time | Usage of GPU Mem |
> | :-: |:-:|:-:|
> | DIMA in Ant 2x4  | ~1d 19h |3.10 GB|
> | MARIE in Ant 2x4 | ~3d 17h |14.33 GB|
> | MAMBA in Ant 2x4 | ~1d 1h  |2.74 GB|
> | DIMA in Ant 4x2  | ~1d 19h |3.15 GB|
> | MARIE in Ant 4x2 | ~3d 17h |15.73 GB|
> | MAMBA in Ant 4x2 | ~1d 1h  |4.37 GB|
>
> **Question 1**: How does the model handle long prediciton horizions? What is the impact of an increasing state/action space? How does the model interpolate or generalize to previously unseen actions/states?
>
> **Response**: Thanks for the insightful questions. We address each of them in turn:
> 1. As our DIMA predicts the next global state based on a fixed window of past global states, it can predict the future over an infinite horizon in an auto-regressive manner.
> 2. With an increasing state and action space, learning an accurate world model becomes more challenging especially in continuous control settings. For example, in the MAMuJoCo suite, each agent has a 2–4 dimensional action space, while in Bi-DexHands, each agent’s action space is 26-dimensional—over 6× larger. This makes modeling the transition dynamics significantly harder. Despite this, DIMA remains robust and performs well in Bi-DexHands, whereas other methods (e.g., MARIE) fail to scale and encounter severe out-of-memory (OOM) issues due to its computational overhead.
> 3. We refer the reviewer to our **response to Weakness 1** for a detailed evaluation. Briefly, we tried to analyze the prediction error accumulation by sampling trajectories using policies from all methods.
> Since learning the world model is tied to a progressively improving policy both in MARIE, MAMBA and DIMA, we separately use their final policies to sample 5 episodes for fairness. We then compute L1 errors per observation between 100 trajectory segments randomly sampled from all 15 episodes and their imagined counterpart. As most of these trajectories are not induced by DIMA’s own policy, they can test how well DIMA generalizes to unseen cases. Results indicate that DIMA significantly outperforms the baselines not only at the training horizon of $H=15$, but also when evaluated beyond the training horizon.
>
>
> **Question 2**: what are the performance of the tested models using more steps / at convergence ?
>
> **Response**: We report the performance of MAPPO, HAPPO and HASAC at 1e7 steps as follows (bold for the highest value, underlined for the second highest). But we would like to emphasize again that **our work focuses on the low data regime**. As shown in Figure 4, our method has not yet fully converged within the original data regime. And we are extending the training of our method to 1e7 steps and would report it as soon as possible.
>
> | Scenarios | MAPPO @ 10M steps | HAPPO @ 10M steps | HASAC @ 10M steps | DIMA @ 1M steps |
> | :-: |:-:|:-:|:-:|:-:|
> | Ant 2x4  |4597 ± 433|4823 ± 775| **5253** ± 617| $\underline{4881}$ ± 756|
> | Ant 4x2  |3375 ± 425|3792 ± 1166| **5338** ± 667| $\underline{4766}$ ± 450|
> | HalfCheetah 6x1 |**7141** ± 180|5453 ± 217|$\underline{6419}$ ± 2564|5643 ± 163|
>
> **Question 3**: What is the computational cost of training the world model (e.g., time), compared to similar approaches?
>
> **Response**: We refer the reviewer to our **response to Weakness 4** for a detailed evaluation.
>
> **Question 4**: Does the authors use “§” to refer to the appendix?
>
> **Response**: Yes, we would add one sentence in the revised manuscript to clarify this symbol usage.
>
>
> **Reference**
>
> [1] Alonso, Eloi, et al. "Diffusion for world modeling: Visual details matter in atari." NIPS (2024): 58757-58791.
>
> [2] Janner, Michael, et al. "When to trust your model: Model-based policy optimization." NIPS (2019).

---

### Comment · Area_Chair_eyu4 · 2025-08-02
**Friendly Reminder: Engaging with Author Rebuttals**

Dear Reviewer,

Thank you for your time and expertise in reviewing for NeurIPS 2025. As we enter the rebuttal phase, we kindly encourage you to:

1) Read and respond to authors' rebuttals at your earliest convenience.

2) Engage constructively with authors by addressing their clarifications in the discussion thread.

3) Update your review with a "Final Justification" reflecting your considered stance post-rebuttal.

Your active participation ensures a fair and collaborative evaluation process. Please don’t hesitate to reach out if you have any questions.

With gratitude,

Your AC

---

### Author Response · Authors · 2025-08-09
**General Response (Part 1)**

Dear PCs, SACs, ACs,

We would like to express our sincere appreciation for the reviewers' time and thoughtful feedback. As the author-reviewer discussion phase comes to a close, we would like to provide a general summary of the reviews and outline the efforts we have made during the rebuttal and discussion phase.

# Summary of Reviews and Responses
Initially, the reviewers deem our proposed formulation and approach as **"novel"**, "conceptually interesting", "well written" with "interesting experiment results", **"clear improvements and advantages in sample efficiency"**, "quite complete empirical analysis" and "extensive experiments".

We summarize the reviewers' concerns and our corresponding responses as follows:

|  | Reviewers' Concerns | Author Responses |
|:-:|:-:|:-:|
| Reviewer Y1VU | Concerns about the scalability of DIMA over longer horizon. | We have conducted additional evaluations with a longer prediction horizon $H = 25$, and evaluated policy learning performance under the extended imagination horizon, comparing DIMA with $H = 15$ and $H = 25$. **Both the results suggest that DIMA not only exhibits lower compounding error trends over time compared to other baselines, but also sustains effective policy learning under a longer imagination horizon, thereby demonstrating its scalability.** |
|| Concerns about the details of the model architectures of DIMA. | We have clarified that the descriptions of our diffusion model architecture are provided in Section G of the appendix.  |
|| Concerns about the performance of baselines with larger data budget. | We have conducted additional experiments and reported the performance of MAPPO, HAPPO and HASAC at 1e7 steps. **When comparing DIMA's performance at 1M steps to strongest model-free baseline HASAC's performance at 5M steps, DIMA achieves higher returns with 5x fewer samples, highlighting its strong sample efficiency.** |
| | Lack of the investigation of the computational cost and training time. | We have measured the training time and GPU memory usage of all evaluated model-based MARL methods. **Overall result shows that DIMA is substantially more efficient than Transformer-based world model (MARIE) in both training time and GPU memory usage, and is comparable to or slightly more efficient than RSSM-based world model (MAMBA).** |
| | Questions about long prediction horizons, scalability with increasing state/action space, and generalization to unseen actions/states. | We have clarified that DIMA predicts the next global state from a fixed window of past states, enabling auto-regressive prediction over an infinite horizon. While larger state/action spaces increase modeling difficulty (e.g., Bi-DexHands’ 26-D action space is 6× larger than MAMuJoCo’s), DIMA remains robust and avoids OOM issues seen in MARIE. For generalization, we evaluate on trajectories not induced by DIMA’s own policy and find it significantly outperforms baselines in prediction accuracy both within and beyond the training horizon. |
| Reviewer DfzB | Concerns about simplicity of the proposed formulation. | We have highlighted that DIMA’s strength lies in its simplicity: it significantly lowers modeling complexity without sacrificing performance by reducing the modeling space. **This principled sequentially conditional denoising formulation enables centralized modeling without explicit communication, yielding strong empirical gains and high sample efficiency.** |
| | Lack of comparison with diffusion-based MARL works MADiff and MADiTs. | We have clarified that both MADiff and MADiTs are offline model-based MARL methods relying on high-quality offline data and goal-conditioned trajectory modeling, whereas DIMA is an online model-based MARL method learned entirely from scratch for direct dynamics modeling. **Due to differences in setting, modeling objective, and usage of diffusion models, they are not directly comparable under our evaluation protocol.** |
| | Suggestions about additional evaluation on benchmarks such as SMACv2. | We have evaluated DIMA across two continuous control benchmarks with 10 scenarios spanning diverse tasks (2–6 agents) where DIMA consistently and significantly outperforms baselines. SMACv2 mainly targets the randomness of agent type and conflicts with DIMA’s implicit fixed-agent-type assumption. Thus we chose not to include the evaluation on SMACv2. |
| | Concerns about the practical implication of Theorem 2. | We have clarified that Theorem 2 theoretically grounds our training objective for sequentially conditioned diffusion-based world models, ensuring the learned transition dynamics align with the intended design. |
| | Missing comparison with recent model-free and Transformer-based MARL methods. | We have clarified that MARIE (Transformer-based world model) is already included. We have additionally compared with HATD3 as a substitute for IMAX-PPO (no open-source code), showing DIMA significantly outperforms it under the same 1M-step low-data regime. |

---

> ### Author Response · Authors · 2025-08-09
> **General Response (Part 2)**
>
> |  | Reviewers' Concerns | Author Responses |
> | :-: |:-:|:-:|
> | | Concerns about the realism of Assumption 1. | We have illustrated with firefighting examples where agents’ actions can sequentially reduce the whole uncertainty in the global state, or reduce the uncertainty in a dimension-wise manner. |
> | | Limitation: need for large datasets to learn accurate dynamics. | We have clarified that unlike offline methods, DIMA learns from scratch (i.e., empty databuffer) in an online setting and achieves strong performance in low-data regimes. |
> | Reviewer Uzmx | Request for numerical results of observation/reward errors over a long horizon. | We have extended prediction horizon from $H=15$ to $H=25$ on Ant-v2-2x4, and reported accumulated observation/reward errors. **Results show DIMA maintains remarkably lower compounding error over time than baselines.** |
> | | Request for clearer comparison about the ablation study. | We have conducted additional experiments on *ShadowHandBottleCap* (Bi-DexHands) with 8 runs. The results demonstrate sequential modeling can show certain advantage over joint under lower data regime (100K/150K steps) in both return and variance. |
> | | Questions on benefits of separately modeling dynamics (diffusion) and reward/termination (transformer). | We have clarified that reward and termination prediction cannot be integrated into the diffusion model due to the difference of the learning objective and using Transformer to model reward function and termination function is inspired by the success of Transformer-based world models. |
> | Reviewer n42q | Concerns about the sample efficiency and lack of training cost comparison. | We have clarified the sample efficiency here means achieving higher performance with fewer environment interactions (low-data regime). **DIMA consistently outperforms baselines in this regime (Fig.4,5, Table 2 in Appendix).**  |
> || Lack of training cost comparison | We have measured the training time and GPU memory usage of all evaluated model-based MARL methods. **Overall result shows that DIMA is substantially more efficient than Transformer-based world model (MARIE) in both training time and GPU memory usage, and is comparable to or slightly more efficient than RSSM-based world model (MAMBA).** |
> | | Concerns on why sequential denoising is preferred over joint modeling. | We have clarified that sequential modeling decompose the exponential joint state-action space into a linearly scaling structure, reducing model complexity and further enabling higher sample efficiency in thelow data regime. 8 experiment runs on ShadowHandBottleCap confirm sequential modeling’s superior returns at 100K and 150K steps (i.e., lower data regime). |
> | | Question on where the proof of Theorem 2 is. | We have clarified that the proof of Theorem 2 is included in the appendix PDF available in the supplementary material. |
>
>
>
> We would like to highlight that, as reviewers Y1VU, DfzB, and n42q have acknowledged, all their concerns have been addressed through our thorough efforts and additional experiments.
> Although Reviewer Uzmx still holds some concerns regarding our ablation study and has provided valuable suggestions for better evaluating the reduced modeling complexity, we appreciate and accept these suggestions and are committed to incorporating them in our further evaluations.
> **More importantly, our proposed DIMA demonstrates a significant and clear improvement in sample efficiency compared to existing model-free and model-based MARL approaches.**
>
> We sincerely appreciate all reviewers’ constructive feedback, and we believe these improvements will help us better communicate the advantages of DIMA to the multi-agent and NIPS community.
>
> Best regards,
>
> Authors

---

### Note · Authors · 2025-08-11

We sincerely thank all reviewers for their constructive feedback. During the rebuttal, we addressed all major concerns with clarifications and new experiments:

- **Scalability**: Extended prediction horizon from $H = 15$ to $H = 25$ shows DIMA sustains low compounding error and effective policy learning, confirming scalability.
- **Sample efficiency**: Compared to strongest model-free baseline HASAC, DIMA achieves higher returns with 5× fewer samples (1M vs. 5M steps). Across 10 diverse tasks, DIMA consistently outperforms model-free and model-based baselines in the low-data regime.
- **Computational cost**: Measured training time and GPU usage show DIMA is substantially more efficient than Transformer-based MARIE, and comparable/slightly better than RSSM-based MAMBA.
- **Ablations**: On *ShadowHandBottleCap*, sequential modeling outperforms joint modeling under low data (100K/150K steps) in both return and variance, supporting our claim that reduced modeling complexity boosts sample efficiency. At larger budgets, performance converges.
- **Model comparisons**: Clarified that MADiff/MADiTs (offline, goal-conditioned) differ fundamentally from our online dynamics modeling, making direct comparison inappropriate. Added HATD3 results as a substitute for IMAX-PPO, with DIMA significantly ahead.
- **Assumptions and theory**: Provided realistic examples supporting Assumption 1, and highlighted Theorem 2’s role in theoretically grounding our objective for sequentially conditioned diffusion models. Proof is included in the appendix.
- **Generalization**: Evaluations on trajectories outside DIMA’s policy distribution show superior prediction accuracy beyond the training horizon.

Reviewers Y1VU, DfzB, and n42q agreed that their concerns were resolved. While Reviewer Uzmx suggested alternative evaluation metrics for modeling complexity, we value this feedback and will incorporate it in future work.

**We encourage the AC and reviewers to refer to our posted General Response (below) for the complete set of detailed clarifications, experimental results, and discussions provided during the rebuttal phase.**

Overall, DIMA offers a principled, simple, and scalable formulation that enables centralized modeling without explicit communication, delivering significant improvements in sample efficiency over state-of-the-art MARL methods. We believe these results, supported by extensive analysis and additional experiments, demonstrate the value of DIMA to the community.

---

### Decision · Program_Chairs · 2025-09-17

**Decision:**

Accept (poster)

**Comment:**

This paper introduces DIMA, a novel diffusion-inspired world model for multi-agent reinforcement learning. The core innovation lies in its sequential prediction of agent actions and state transitions, which reduces modeling complexity by aligning with the reverse steps of a diffusion process.

The paper was praised for its exceptional clarity and the strong empirical performance of DIMA on established MARL benchmarks. Reviewers unanimously agreed on the novelty of its sequential, diffusion-inspired modeling approach. The primary concerns raised during review involved the method's scalability over longer prediction horizons, the depth of comparison to relevant prior works, and the need for more rigorous ablation studies to conclusively validate the benefits of sequential modeling over joint alternatives. The authors' rebuttal was exceptionally thorough and directly addressed the majority of these concerns, successfully satisfying three of the four reviewers. However, one reviewer (R3) remained unsatisfied, particularly regarding the sufficiency of the ablation studies. The final scores were three "Weak Accept" and one "Weak Reject."

Given the paper's clear novelty, strong results, and the authors' comprehensive response, I recommend Acceptance. To fully address the residual concerns and solidify the paper's contribution, I strongly urge the authors to incorporate the following into their camera-ready version: additional experiments comparing observation and reward prediction errors for sequential vs. joint modeling. The addition will significantly enhance the paper's rigor and reproducibility for the community.